# Giant ankyrin-B mediates transduction of axon guidance and collateral branch pruning factor sema 3A

**Blake A Creighton[1†], Simone Afriyie[1†], Deepa Ajit[1], Cristine R Casingal[1,2], Kayleigh M Voos[1], Joan Reger[3,4], April M Burch[1], Eric Dyne[3], Julia Bay[1], Jeffrey K Huang[4], ES Anton[1,2], Meng-Meng Fu[3], Damaris N Lorenzo[1,2,5]\***

[1]Department of Cell Biology and Physiology, University of North Carolina at Chapel Hill, Chapel Hill, United States; [2]Neuroscience Center, University of North Carolina at Chapel Hill, Chapel Hill, United States; [3]National Institute of Neurological Disorders and Stroke (NINDS), Bethesda, United States; [4]Department of Biology and Center for Cell Reprogramming, Georgetown University, Washington, United States; [5]Carolina Institute for Developmental Disabilities, Chapel Hill, United States

**\*For correspondence:**
damaris_lorenzo@med.unc.edu

[†]These authors contributed equally to this work

**Abstract** Variants in the high confident autism spectrum disorder (ASD) gene *ANK2* target both ubiquitously expressed 220 kDa ankyrin-B and neurospecific 440 kDa ankyrin-B (AnkB440) isoforms. Previous work showed that knock-in mice expressing an ASD-linked *Ank2* variant yielding a truncated AnkB440 product exhibit ectopic brain connectivity and behavioral abnormalities. Expression of this variant or loss of AnkB440 caused axonal hyperbranching in vitro, which implicated AnkB440 microtubule bundling activity in suppressing collateral branch formation. Leveraging multiple mouse models, cellular assays, and live microscopy, we show that AnkB440 also modulates axon collateral branching stochastically by reducing the number of F-actin-rich branch initiation points. Additionally, we show that AnkB440 enables growth cone (GC) collapse in response to chemorepellent factor semaphorin 3 A (Sema 3 A) by stabilizing its receptor complex L1 cell adhesion molecule/neuropilin-1. ASD-linked *ANK2* variants failed to rescue Sema 3A-induced GC collapse. We propose that impaired response to repellent cues due to AnkB440 deficits leads to axonal targeting and branch pruning defects and may contribute to the pathogenicity of *ANK2* variants.

## Editor's evaluation

In their paper, Creighton et al. investigate the mechanisms regulating cytoskeletal changes mediating neuronal branching and axon growth. They assess the role of the neuronal specific form of the scaffolding protein ankyrin-B, and how its depletion or autism-spectrum disorder mutations affect cortical axon branching, targeting, and developmental refinement. Overall, their work develops a new model for how ankyrin-B functions with the Sema 3A receptor complex formed by cell adhesion molecule L1CAM and neuropilin-1, and an actin severing protein (cofilin) during cortical axon development and provides insight into how this process is altered in autism spectrum disorders. This manuscript will be of interest to cell, developmental, and neuronal biologists interested in the biological mechanisms of early circuit development, and how these mechanisms relate to autism spectrum disorders.

## Introduction

Precise wiring of the neural circuitry relies on a tightly regulated spatiotemporal program of axonal extension and pathfinding to form synapses with their targets. The remarkable task of correctly extending the axon through dense three-dimensional spaces such as the developing mammalian brain

is driven by the growth cone (GC), the highly dynamic structure that provides the mechanical force at the tip of the growing axon. To contact targets not in their path and to maximize the number of synapses, most neurons multiply interneuron contact points through collateral axon branches, which are also extended by individual GCs (*Kalil and Dent, 2014*; *Armijo-Weingart and Gallo, 2017*). In humans, most synapses are assembled during brain development, which number is markedly reduced by adulthood via refinements that include the pruning of axons, collateral branches, dendrites, and dendritic spines (*Riccomagno and Kolodkin, 2015*; *Low and Cheng, 2006*; *Petanjek et al., 2011*; *Moriyama et al., 1996*). The growth and guidance of axonal processes and the sculpting of synaptic connections are determined by multiple intrinsic and extrinsic cellular factors (*Riccomagno and Kolodkin, 2015*). Mistargeting, deficits or excess of axonal processes can lead to hypo-, hyper-, or aberrant neuronal connectivity, which in turn may underlie functional deficits associated with psychiatric and neurodevelopmental disorders (*Holiga et al., 2019*; *Taquet et al., 2020*).

Ankyrin-B (AnkB) is an integral component of the submembrane cytoskeleton involved in the organization of specialized domains that include ion channels, membranes transporters, cell adhesion molecules, and other scaffolding proteins (*Bennett and Lorenzo, 2013*; *Bennett and Lorenzo, 2016*). AnkB has emerged as a critical determinant of structural neural connectivity through diverse and divergent roles of its two major, alternatively spliced isoforms in the brain; neuronal-specific 440 kDa ('giant') AnkB (AnkB440) and ubiquitously expressed 220 kDa AnkB (AnkB220) (*Lorenzo, 2020*; *Figure 1A*). AnkB knockout mice lacking expression of both AnkB220 and AnkB440 exhibit noticeable reduction in the length and number of long axon tracts in multiple brain areas and experience early neonatal mortality (*Scotland et al., 1998*; *Lorenzo et al., 2014*). Correspondingly, cultured neurons from these mice fail to properly elongate their axons, in part due to reductions in the axonal transport of multiple organelles. AnkB220 enables organelle transport by coupling the retrograde motor complex to vesicles in axons (*Lorenzo et al., 2014*) and other cell types (*Qu et al., 2016*; *Lorenzo et al., 2015*; *Lorenzo and Bennett, 2017*). Rescue with AnkB220 is sufficient to restore both normal axonal growth and organelle trafficking in vitro. In contrast, the lifespan and overall development of mice lacking only AnkB440 (AnkB440 KO) appear normal. However, AnkB440 KO mice exhibit abnormal communication and sexual behavior responses, repetitive behavior, and increased executive function in the absence of ASD-associated comorbidities (*Yang et al., 2019*). These behavioral deficits are also present in knock-in mice bearing a frameshift mutation (AnkB440$^{R2589fs}$) in the inserted exon unique to AnkB440 that models a human variant found in an individual diagnosed with ASD. AnkB440$^{R2589fs}$ mice have normal AnkB220 expression in the brain, and instead of full-length AnkB440, they express reduced levels of a truncated 290 kDa AnkB polypeptide lacking a portion of the neuronal-specific domain (NSD) and the entire death and C-terminal regulatory domains (*Yang et al., 2019*). Diffusion tension imaging (DTI) tractography of AnkB440$^{R2589fs}$ brains revealed the presence of ectopic axon projections and stochastic alterations in axonal connectivity, consistent with a potential role of AnkB440 in modulating the formation, guidance, and refinement of axonal connections in the developing brain. In line with these findings, primary cortical neurons from both AnkB440 KO and AnkB440$^{R2589fs}$ mice exhibit increased axonal branching (*Yang et al., 2019*). AnkB440 directly binds and bundles microtubules through a bipartite microtubule-interaction site within the neuronal-specific domain, which is required to suppress ectopic axonal branching in cultured neurons (*Chen et al., 2020*). Furthermore, AnkB440, but not AnkB220, binds L1 cell adhesion molecule (L1CAM) at axonal membranes, where AnkB440 and L1CAM collaborate to coordinate positioning and local dynamics of cortical microtubules and prevent the stabilization of nascent axonal filopodia and maturation into collateral axon branches. In agreement with this model, cultured neurons from knock-in mice bearing the L1CAM variant p.Y1229H, which lacks AnkB440 binding activity, show excess axon collateral branching (*Yang et al., 2019*).

Here, we show that AnkB440 also modulates axon collateral branching stochastically by suppressing the formation of F-actin-rich branch initiation points. We demonstrate that axons require AnkB440, but not AnkB220, to transduce the chemorepellent signals induced by the extrinsic factor semaphorin-3A (Sema 3 A). GCs in the primary axon and in collateral branches of cultured cortical neurons from AnkB440 KO mice fail to collapse in response to Sema 3 A. This suggests that in addition to suppressing axon collateral branch initiation, AnkB440 is involved in collateral branch pruning. However, unlike the suppression of branch point initiation (*Chen et al., 2020*), GC collapse and branch retraction did not involve the interaction of AnkB440 with microtubules. On the other hand, binding

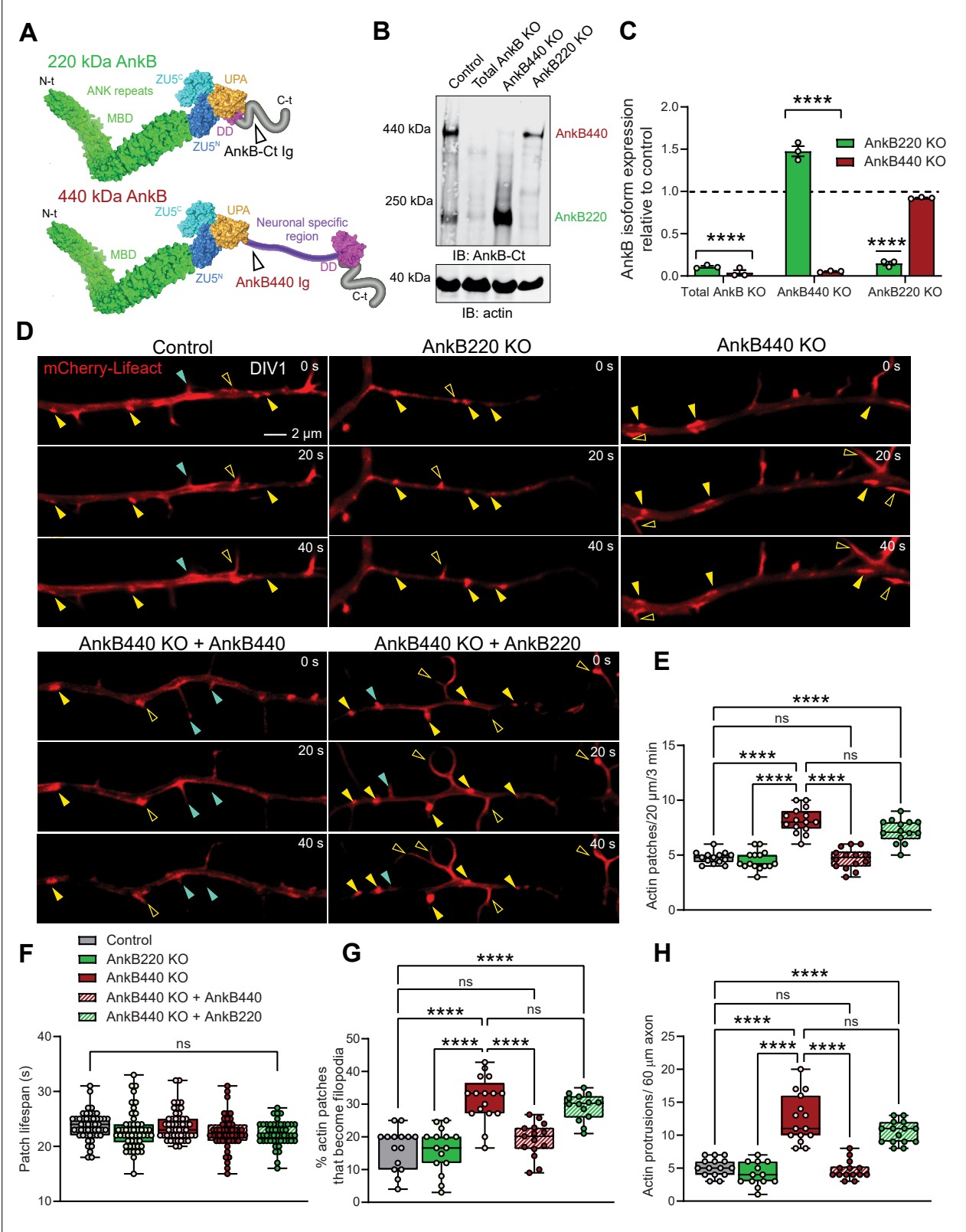

**Figure 1.** Ankyrin-B 440 suppresses the formation of axonal filopodia in vitro. (**A**) Representation of the two major ankyrin-B isoforms in the mammalian brain and their functional domains. The localization of the epitopes recognized by the pan anti-ankyrin B (AnkB-Ct Ig) or the anti-440 kDa ankyrin-B (AnkB440 Ig) antibodies is indicated by arrowheads. (**B**) Western blot analysis of expression of ankyrin-B isoforms in the cortex of PND1 control mice and of mice individually lacking the 220 kDa ankyrin-B isoform (AnkB220 KO), the 440 kDa ankyrin-B isoform (AnkB440 KO), and both isoforms (Total AnkB

*Figure 1 continued on next page*

*Figure 1 continued*

KO). (**C**) Quantification of AnkB220 and AnkB440 levels normalized to actin in cortical lysates from PND1 mice of indicated genotypes relative to their levels in control brains. Data show mean ± SEM for three biological replicates per genotype representative of one of three independent experiments. Unpaired *t* test. ****$p < 0.0001$. (**D**) Images selected from timelapse sequences of DIV1 neurons expressing mCherry-Lifeact showing the formation of actin patches (asterisks), some of which do not form filopodia (closed yellow arrowheads) or lead to filopodia that may persist (open yellow arrowheads) or retract (closed teal arrowheads). Scale bar, 2 µm. (**E**) Quantification of actin patches observed in 20 µm of axons during a 3 minute acquisition period. Data was collected from n = 14 control, n = 15 AnkB220 KO, n = 15 AnkB440 KO, n = 14 AnkB440 KO+ AnkB440, and n = 14 AnkB440 KO+ AnkB220 axons from three independent experiments. (**F**) Quantification of actin patch lifespan derived from the analysis of n = 50 patches per each group collected from three independent experiments. (**G**) Quantification of proportion of actin patches that result in filopodia during the movie acquisition interval derived from analyses of n = 15 control, n = 15 AnkB220 KO, n = 16 AnkB440 KO, n = 15 AnkB440 KO+ AnkB440, and n = 14 AnkB440 KO+ AnkB220 axons from three independent experiments. (**H**) Quantification of filopodia per 60 µm of axons obtained from the analysis of n = 15 control, n = 13 AnkB220 KO, n = 15 AnkB440 KO, n = 14 AnkB440 KO+ AnkB440, and n = 15 AnkB440 KO+ AnkB220 axons from three independent experiments. The box and whisker plots in **E–H** represent all data points collected arranged from minimum to maximum. One-way ANOVA with Tukey's post hoc analysis test for multiple comparisons. $^{ns}p > 0.05$, ****$p < 0.0001$.

The online version of this article includes the following video and figure supplement(s) for figure 1:

**Source data 1.** Ankyrin-B 440 regulates the formation of axonal filopodia and branches.

**Figure supplement 1.** Development of conditional AnkB220 knockout mouse.

**Figure supplement 2.** AnkB440 loss increases axon filopodia without early changes in total neurite count.

**Figure supplement 3.** AnkB440 and AnkB220 isoforms exert different roles in axonal growth and branching.

**Figure supplement 4.** AnkB220, but not AnkB440, promotes axonal organelle transport.

**Figure 1—video 1.** AnkB220 promotes axonal organelle transport.

https://elifesciences.org/articles/69815/figures#fig1video1

activity between AnkB440 and L1CAM is necessary for normal transduction of Sema 3 A cues and for downstream actin cytoskeleton remodeling, but it does not involve AnkB partner βII-spectrin. We propose that this mechanism contributes to defects in the formation, organization, and targeting of axonal tracts observed in AnkB440 KO mouse brains. Lastly, we found that selected *ANK2* variants found in individuals with ASD phenocopied the loss of response to Sema 3 A, indicating a potential mechanism of pathogenicity.

## Results

### Loss of AnkB440, but not AnkB220, results in higher density of actin patches, filopodia, and collateral axon branches

Most collateral branches emerge from transient actin-rich nucleation sites along the axon, which give rise to branch precursor filopodia (*Gallo, 2011*). Commitment to branch formation requires microtubule invasion of the nascent filopodia (*Yu et al., 1994*; *Kalil and Dent, 2014*). AnkB440 recruits and stabilizes microtubules at the plasma membrane and its loss has been proposed to cause axonal hyperbranching in vitro by promoting microtubule unbundling and invasion of nascent branches (*Yang et al., 2019*; *Chen et al., 2020*). AnkB440 loss results in 50 % upregulation of AnkB220 in brains of AnkB440 KO mice (*Yang et al., 2019*; *Figure 1B and C*; see *Figure 1—source data 1*). We previously reported that AnkB220 promotes axonal growth and axonal transport of organelles, including mitochondria (*Lorenzo et al., 2014*), which has been implicated in axonal hyperbranching (*Courchet et al., 2013*). Likewise, AnkB220 associates with microtubules in biochemical assays (*Davis and Bennett, 1984*). Changes in AnkB220 expression could contribute to or compensate for deficits in AnkB440 and its effects on axonal branching. Thus, we sought to determine whether AnkB220 and AnkB440 have independent roles in axonal branch formation using cortical neurons harvested from AnkB440 KO mice (*Yang et al., 2019*; *Chen et al., 2020*) and from a novel conditional AnkB220 KO mouse model (herein referred as AnkB220 KO) (*Figure 1B*, see *Figure 1—source data 1*). The conditional *Ank2* allele used to selectively eliminate AnkB220 expression was generated by fusing exons 36 and 37 of *Ank2* to prevent splicing of exon 37 unique to the AnkB440 transcript (*Figure 1—figure supplement 1A*). Western blot analysis from cortical lysates of mice in which recombination of the *Ank2* floxed allele (*Ank2*$^{220flox}$) (*Figure 1—figure supplement 1B*, see *Figure 1—source data 1*) was

induced by Nestin-Cre (*Ank2$^{220flox/220flox}$/Nestin-Cre*), referred to as AnkB220 KO, demonstrates efficient knockdown of AnkB220 and preservation of AnkB440 expression (*Figure 1A and B*).

The focal accumulation of dynamic pools of F-actin, termed actin patches, along the axon shaft has been shown to promote filopodia formation, the first step during collateral axon development (*Gallo, 2011*; *Spillane et al., 2011*; *Armijo-Weingart and Gallo, 2017*). Furthermore, the presence of larger and more stable actin patches within the primary axon directly correlates with the formation of axon protrusions in vivo (*Hand et al., 2015*). We found that loss of AnkB440 in day in vitro 1 (DIV1) cortical neurons resulted in increased number of actin patches relative to control and AnkB220 KO neurons (*Figure 1D and E*, see *Figure 1—source data 1*). These mCherry-Lifeact-labeled F-actin patches marked potential sites for emerging filopodia, some of which failed to initiate filopodia extension (*Figure 1D, closed yellow arrowheads*), while others extended filopodia (*Figure 1D, open yellow arrowheads*), a portion of which retracted and vanished (*Figure 1D, closed teal arrowheads*). While the lifespan of actin patches was similar across genotypes (*Figure 1F*), a higher proportion of actin patches developed into nascent filopodia protrusions in AnkB440 KO neurons (*Figure 1G*), which correlated with a higher density of both axonal filopodia (*Figure 1H* and *Figure 1—figure supplement 2A*, see *Figure 1—source data 1*) and collateral axon branches (*Figure 1—figure supplement 3A,B*, see *Figure 1—source data 1*). Neurons lacking AnkB440 extended similar number of processes as control neurons prior to axon specification (*Figure 1—figure supplement 2B,C*). However, loss of AnkB220, but not of AnkB440, resulted in shorter axons (*Figure 1—figure supplement 3A,C,D*) and impaired organelle transport (*Figure 1—figure supplement 4A-D* and *Figure 1—video 1*, see *Figure 1—source data 1*), which confirms our previous findings based on in vitro rescue experiments that AnkB220 is critical for axonal growth and organelle dynamics (*Lorenzo et al., 2014*). Deficits in actin patch formation, axon filopodia density, and number of collateral axon branches of AnkB440 KO neurons were restored by expression of AnkB440, but not of AnkB220 (*Figure 1D–H* and *Figure 1—figure supplement Figure 1—figure supplements 2A and 3A-D*). This indicates that while AnkB440 suppresses collateral branch formation, AnkB220 is not a cell-autonomous determinant of axon branching, and that elevations in AnkB220 do not influence organelle dynamics in AnkB440 deficient axons. Together, our data suggest that in addition to the previously proposed role of loss of AnkB440 in stabilizing axon branching by promoting microtubule invasion of nascent axonal filopodia (*Yang et al., 2019*), AnkB440 deficits also contribute to axonal hyperbranching stochastically by inducing the formation of a higher number of axonal filopodia.

## Loss of AnkB440 results in increased axonal collateral branching in vivo and altered development of axonal tracts

DTI tractography analysis of AnkB440$^{R2589fs}$ brains, which lack full-length AnkB440 but express a truncated 290 kDa AnkB440 polypeptide, display stochastic increases in cortical axon connectivity in vivo (*Yang et al., 2019*). Unlike AnkB440 that preferentially localizes to axons, the truncated fragment is equally distributed between dendrites and axons (*Yang et al., 2019*). The changes in distribution of this AnkB440 fragment, which retains some of the domains and functions of AnkB440, could lead to gain-of-function or dominant negative effects that may differentially influence structural or functional axonal connectivity relative to total AnkB440 loss.

To determine whether AnkB440 loss alters axon connectivity in vivo, we first visualized projection neurons from the upper layers of the primary somatosensory cortex of postnatal day 25 (PND25) brains using Golgi staining. In agreement with the in vitro results, AnkB440 KO axons traced from brain slices had on average four times more branches in the axon region proximal to the soma than both AnkB220 KO and control axons (*Figure 1—figure supplement 3E,F*). Next, we assessed the morphology and trajectory of axon tracts in brains of PND25 control and AnkB440 KO mice, and of mice of the same genotypes that also express a green fluorescent protein (GFP) reporter (Thy1::G-FP–M mouse line) (*Feng et al., 2000*) in a subset of neurons, which allows for sparse labeling and tracing of selective axon fibers. AnkB440 KO mice exhibited corpus callosum (CC) hyperplasia (*Figure 2A–C* and *Figure 2—figure supplement 1A,B*, see *Figure 2—source data 1*) that was not due to due to changes in the abundance and distribution of callosal projection neurons in neocortical layers, assessed by expression of the transcription factors Satb2, Ctip2, and Tbr1 (*Fame et al., 2011*; *Figure 2—figure supplement 1C,D*). AnkB440 KO brains also displayed volumetric changes in the anterior commissure (AC), the ventral hippocampal commissure (VHC), and the septofimbrial nucleus

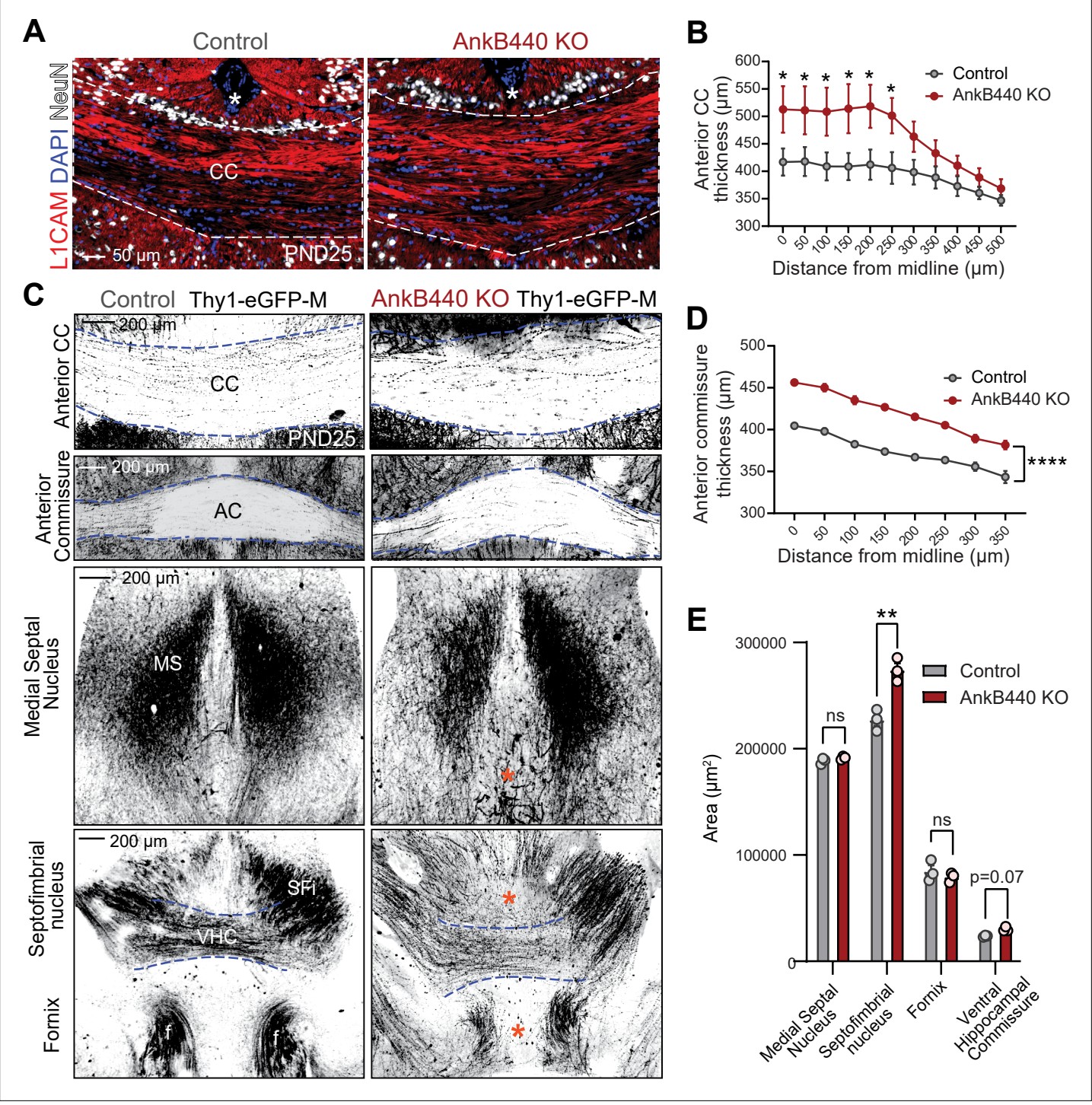

**Figure 2.** Loss of AnkB440 alters the formation of axonal tracts in mouse brains. (**A**) Images of coronal sections of PND25 mouse brains taken through the anterior portion of the corpus callosum (CC) and stained for L1CAM to label axons, and DAPI and NeuN to respectively stain total and neuronal nuclei. An asterisk indicates the position of the brain midline. Scale bar, 50 µm. (**B**) Quantification of thickness of anterior region of the CC at different points from the brain midline from control (n = 4) and AnkB440 KO (n = 5) brains. (**C**) Images of different axon tracts of control and AnkB440 KO PN25 brains that also express the Thy1-GFP-M reporter. Scale bar, 200 µm. Blue lines demark the boundaries of indicated commissural tracts. Orange asterisks indicate mistargeted axons. AC (anterior commissure), MS (medial septal nucleus), VHC (ventral hippocampal commissure), SFi (septobimbrial nucleus). (**D**) Quantification of thickness of the AC at different points from the brain midline from control (n = 3) and AnkB440 KO (n = 3) brains. (**E**) Area of indicated tracts evaluated from control (n = 3) and AnkB440 KO (n = 3) Thy1-GFP-M-positive brains. Data in **B**, **D**, **E** represent mean ± SEM. Unpaired $t$ test. $^{ns}p > 0.05$, $^*p < 0.05$, $^{**}p < 0.01$, $^{****}p < 0.0001$ (for all distances evaluated in **D**).

*Figure 2 continued on next page*

*Figure 2 continued*

The online version of this article includes the following figure supplement(s) for figure 2:

**Source data 1.** Loss of AnkB440 alters axonal tracts in mouse brains.

**Figure supplement 1.** AnkB440 loss causes corpus callosum hyperplasia.

(SFi) (*Figure 2C–E*). L1CAM and GFP labeling of axons also detected changes in axon organization within the tracts analyzed, including axon defasciculation and the presence of ectopic or misrouted axons (*Figure 2A and C*, *blue asterisk*). Taken together, these histological evaluations reveal abnormalities in the establishment, organization, and routing of forebrain axon fibers in AnkB440 KO brains, likely involving axon branching, fasciculation, guidance, and pruning pathways.

CC hyperplasia can result from changes in axon myelination due to alterations in oligodendrocyte development through both intrinsic and non-cell-autonomous mechanisms (*Wang et al., 2012*; *Mitew et al., 2018*). To assess a potential direct effect of AnkB440 loss on oligodendrocytes, we first evaluated its expression in differentiated rat oligodendrocyte progenitor cells (OPCs). Staining with anti-AnkB440 antibody revealed a weak signal around the nucleus and in small puncta around the processes, which either did not change or had very modest intensity differences with OPC maturation, assessed by morphology and increased myelin basic protein (MBP) staining (*Figure 3—figure supplement 1A,B*). We observed AnkB220 but not AnkB440 expression in protein lysates from astrocyte or DIV3 OPC cultures (*Figure 3—figure supplement 1C*; see *Figure 3—source data 1*). Levels of AnkB440 transcripts were also negligible in DIV3 OPCs, a stage at which *Mbp* mRNA is actively transported but not yet locally translated (*Figure 3—figure supplement 1D*). These findings rule out a cell-autonomous effect of AnkB440 loss in oligodendrocytes.

AnkB440 loss resulted in lower Olig2$^+$ oligodendrocyte density in the CC (*Figure 3—figure supplement 2A-C*, see *Figure 3—source data 1*) but did not affect the density or g-ratio of myelinated axons (*Figure 3A–E*), as assessed by transmission electron microscopy (TEM), or the levels of MBP in the CC (*Figure 3F and G*). Expression of mature oligodendrocyte-specific proteins in cortical lysates of AnkB440 KO mice was also unchanged relative to control (*Figure 3H, I*; see *Figure 3—source data 1*). On the other hand, electron micrographs of AnkB440 KO CC axons showed widespread lose wraps in the myelin sheath near the axonal membrane (*Figure 3A*, *open red arrowhead*) and enlarged inner tongues, the space inside the myelin sheath adjacent to the axon (*Figure 3A*, *open blue arrowheads*). Thus, AnkB440 is likely involved in contact-mediated signals between the axon and myelin wrap that promote myelin ultrastructural organization and health, which could impact CC axon organization and function when defective.

## AnkB440 deficiency alters the establishment and refinement of callosal projections

To further assess the effects of AnkB440 deficit on axon hyperbranching and its contribution to the establishment and refinement of callosal projections in vivo, we expressed GFP in layer II/III neurons of the somatosensory cortex region S1 by in utero electroporation (IUE) of E15.5 embryos (*Figure 4A*). First, we evaluated the distribution of GFP-labeled axons at the CC midline at PND25. In control brains, most GFP-labeled S1 callosal axons crossed through the ventral portion of the CC (60–100 μm from the most dorsal CC region), with only a few axons distributed throughout the dorsal CC regions (*Figure 4B insets 1*, *C*) (*Zhou et al., 2013*). This organized CC axon topography is achieved between PND16 and PND30 through developmental pruning of ectopically localized dorsal S1 CC axons (*Zhou et al., 2013*; *Martín-Fernández et al., 2021*). GFP-labeled CC axons showed a more widespread distribution at the midline of PND25 AnkB440 KO brains, with a significant number of axons navigating through more dorsal CC paths (30–60 μm from the most dorsal CC region) (*Figure 4B* insets 3, C, see *Figure 4—source data 1*). As expected, GFP-labeled control callosal axons reached their targets in the corresponding contralateral S1/S2 border region, where they branched extensively and preferentially innervated cortical layers II/III and V (*Figure 4B inset 2,D*) (*Fenlon et al., 2017*). In contrast, this pattern of homotopic axon targeting to the S1/S2 border was lost in AnkB440 KO brains. Instead, GFP-labeled S1 projections were distributed throughout the contralateral somatosensory cortex, with a notable fraction of axons ectopically projecting to more medial S1 regions and to regions below the S1/S2 border (*Figure 4B inset 4,D*). AnkB440 KO brains also showed more contralateral axon

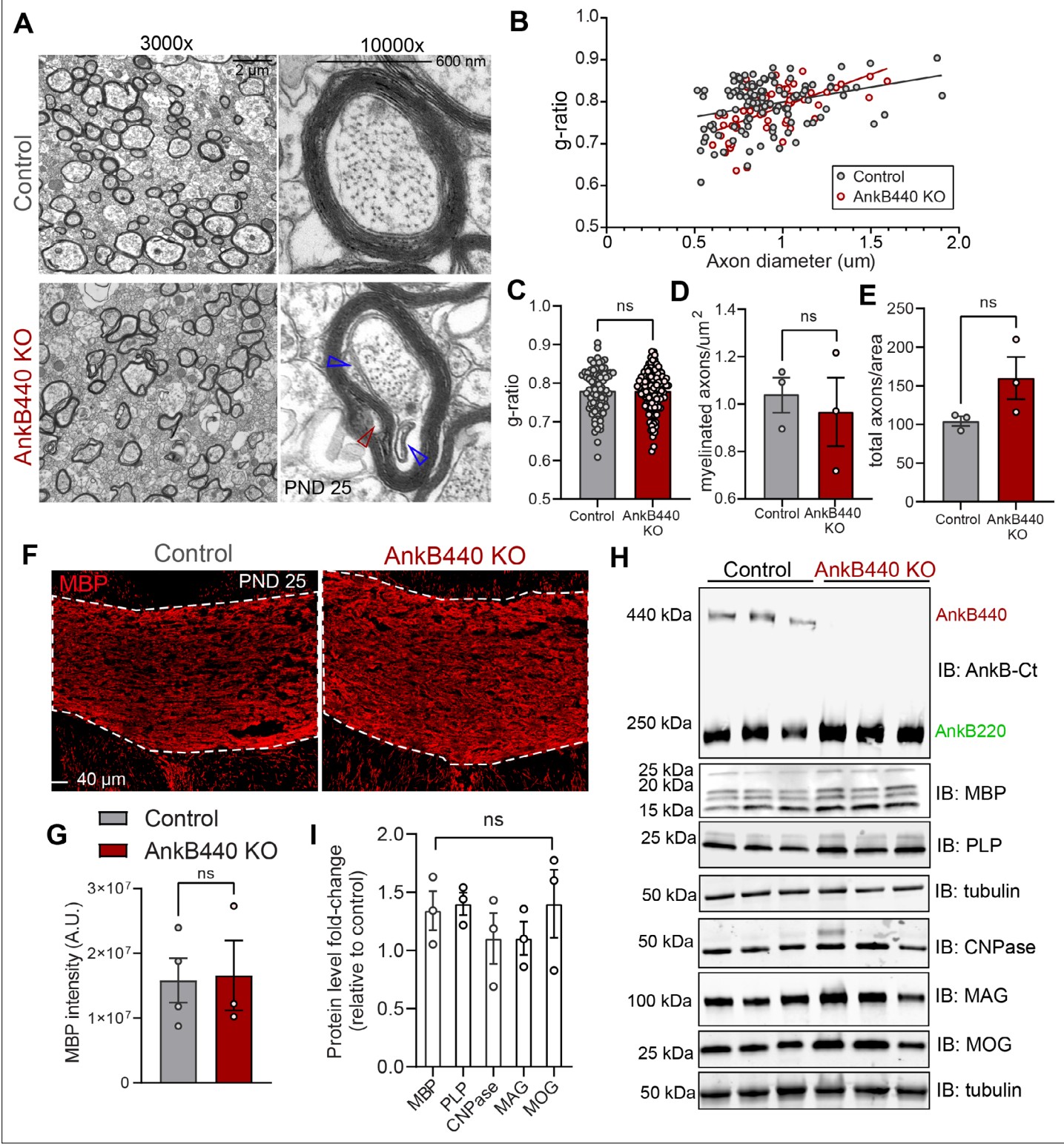

**Figure 3.** Loss of AnkB440 does not change levels of myelination of CC axons. (**A**) TEM images of cross-sections through the CC near the brain midline from PND25 control and AnkB440 KO mice (n = 3 mice/genotype). Images were taken at 3000 x (scale bar, 2 μm) and 10,000 x (scale bar, 600 nm) magnification. Open red and blue arrowheads respectively indicate loose myelin wraps and enlarged inner tongues. (**B**) G-ratio versus axon diameter computed from control (n = 115) and AnkB440 KO (n = 63) axons from n = 3 mice/genotype. (**C**) Myelin g-ratio computed from (n = 84) and AnkB440 KO (n = 120) axons from n = 3 mice/genotype. (**D**) Myelinated axons per μm² and (**E**) total axons per field of view (10,000 x images) computed from at least three independent images per mouse and n = 3 mice/genotype. (**F**) Images of coronal sections of PND25 mouse brains taken through the anterior

*Figure 3 continued*

portions of the CC (white lines) and stained for MBP. Scale bar, 40 μm. (**G**) MBP fluorescent intensity in the CC computed from n = 3 brains/genotype. (**H**) Western blot analysis of the expression of AnkB isoforms and multiple proteins (MBP, myelin-PLP, CNPase, MAG, MOG) associated with myelination in the cortex of PND25 control and AnkB440 KO mice (n = 3 lysates/genotype). (**I**) Fold-change in levels of myelination-associated proteins in cortical lysates from PND25 AnkB440 KO mice normalized to tubulin and computed relative to the expression of each protein in control brains. Data in **C–E**, **G**, and **I** show mean ± SEM for three biological replicates per genotype. Unpaired *t* test. $^{ns}$p >0.05.

The online version of this article includes the following figure supplement(s) for figure 3:

**Source data 1.** Loss of AnkB440 does not alter levels of myelination of CC axons.

**Figure supplement 1.** AnkB440 is not expressed at detectable levels in differentiated OPCs.

**Figure supplement 2.** AnkB440 KO mice have normal number of oligodendrocytes in the corpus callosum.

branches aberrantly innervating all cortical layers (*Figure 4B and E*). Together, these findings identify aberrant topographic organization and targeting of callosal projections in AnkB440 KO mice that are not corrected by developmental pruning processes.

## AnkB440 is required for normal growth cone collapse in response to semaphorin 3A

AnkB440 interacts with L1CAM (*Yang et al., 2019*), which is required for transducing the repulsive growth and guidance effects of the soluble guidance cue class III semaphorin A (Sema 3 A) (*Castellani et al., 2000*; *Castellani et al., 2002*; *Castellani et al., 2004*) through its direct binding to the Sema 3 A receptor neuropilin 1 (Nrp1) (*Chen et al., 1998*; *Kolodkin et al., 1997*). Sema 3 A promotes GC collapse (*Luo et al., 1993*; *Kolodkin et al., 1993*) and inhibits axon branching in vitro (*Dent et al., 2004*), which likely underlies its roles in repelling cortical axons and in pruning axonal branches in vivo and in vitro (*Polleux et al., 1998*; *Bagri et al., 2003*). Nrp1-mediated Sema 3 A signaling also promotes the topographic segregation of axons from different cortical regions within the CC, including the ventral localization of somatosensory S1 region callosal axons, which, in turn, determines the establishment of homotopic contralateral projections (*Zhou et al., 2013*; *Martín-Fernández et al., 2021*).

Previous studies suggest that AnkB440 preferentially localizes to axons (*Chan et al., 1993*; *Yang et al., 2019*; *Chen et al., 2020*). Staining of DIV3 cortical neurons with an antibody specific for AnkB440 (*Yang et al., 2019*; *Figure 5—figure supplement 1A*, see *Figure 5—source data 1*) confirmed the widespread distribution of AnkB440 in axons, with strong AnkB440 signals also detected at the tips of axonal process, including nascent filopodia, and at the GCs of the primary axon and collateral branches (*Figure 5—figure supplement 1B,C*). As expected, AnkB440 signal was absent from axons of AnkB440 KO neurons stained with the AnkB440-specific antibody (*Figure 5—figure supplement 1D,i*). Neither the overall axonal distribution nor the localization of AnkB440 at GCs changed in neurons selectively lacking AnkB220 stained with AnkB440-specific (*Figure 5—figure supplement 1D,ii*) or pan-AnkB (*Figure 5—figure supplement 1D,iii*) antibodies. Interestingly, AnkB440 localized to both microtubule- and F-actin-enriched axonal domains (*Figure 5—figure supplement 1C,D*), stained with βIII-tubulin and phalloidin respectively, suggesting that it might promote the organization and/or dynamics of both cytoskeletal networks.

Given the abundant expression of AnkB440 in GCs of cortical neurons together with the similarities in disruption of the topographic order of midline S1 callosal axons and in the presence of heterotopic contralateral callosal projections among AnkB440 KO (*Figure 4*), Sema 3 A KO, and Nrp1 KO brains (*Zhou et al., 2013*; *Martín-Fernández et al., 2021*), we used a GC collapse assay to evaluate whether cortical neurons require AnkB440 to transduce Sema 3 A cues. Roughly 60 % of GCs of both control and AnkB220 KO cortical neurons collapsed after 30 min of Sema 3 A treatment, up from a baseline of 20 % GC collapse in untreated cells, without any appreciable difference between the two groups (*Figure 5A and B*). In contrast, GCs of total AnkB KO and AnkB440 KO neurons significantly failed to collapse when exposed to Sema 3 A (*Figure 5A, C and D*, see *Figure 5—source data 1*). The failure of GCs to collapse upon Sema 3 A treatment in AnkB440 KO neurons was restored by transfection with AnkB440 but not with AnkB220 plasmids (*Figure 5E–G*). To rule out that the inability of Sema 3 A to collapse GCs in AnkB440 KO neurons was due to a generalized loss of transduction of chemorepulsive cues, we treated control and AnkB440 neurons with Ephrin A5, which also induces GC collapse in cortical neurons (*Meima et al., 1997*). Ephrin A5 promoted GC collapse in AnkB440 KO cultured

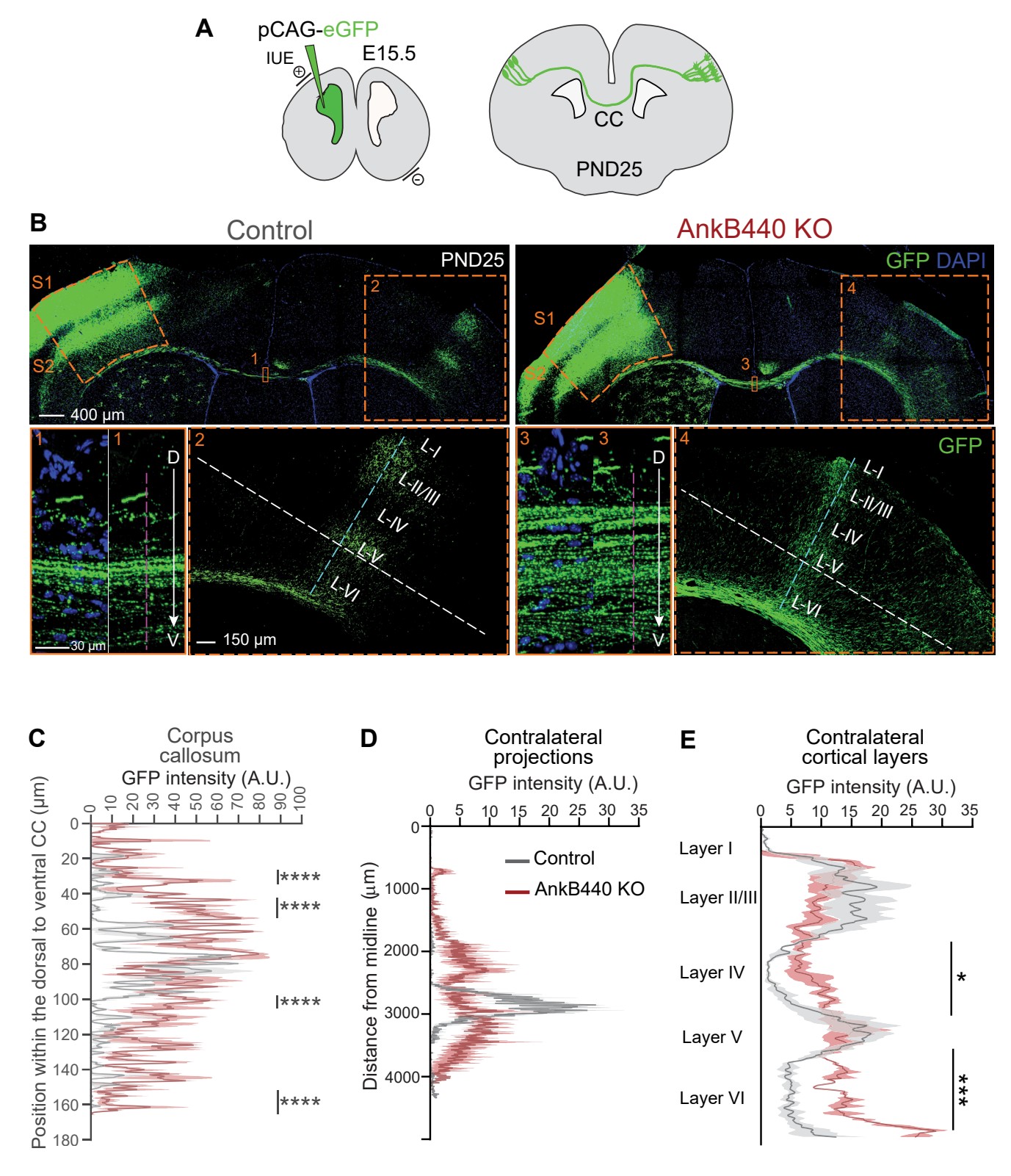

**Figure 4.** AnkB440 deficiency alters the establishment and refinement of callosal projections. (**A**) Diagram illustrating the strategy used to label callosal projecting axons with GFP via IUE of layer II/III S1 neurons at E15.5 and to assess their contralateral targeting at PND25. (**B**) Coronal sections of PND25 control and AnkB440 KO brains electroporated with pCAG-GFP at E15.5. Scale bar, 400 μm. Insets 1 and 3 (scale bar, 30 μm) show high-magnification images of the boxed CC regions. Insets 2 and 4 (scale bar, 150 μm) show high magnification images of the boxed contralateral cortical regions.

*Figure 4 continued on next page*

*Figure 4 continued*

The dorsoventral (D–V) order of S1 axons within the CC was grossly disrupted in AnkB440 KO mice (inset 3). (**C**) Quantification of the dorsoventral distribution of GFP signal in the CC at PND25 as a function of the distance from the most dorsal (**D**) portion of the CC, measured through the region demarked by a discontinuous pink line in insets 1 and 3. (**D**) GFP labeling reveals disruption of homotopic S1 callosal projection targeting to the S1/S2 border in AnkB440 KO brains (inset 4), and more contralateral axon branches aberrantly innervating all cortical layers. (**D**) Quantification of GFP fluorescence of contralateral axon projections as a function of their distance from the brain midline measured through the region demarked by a discontinuous white line in insets 2 and 4. (**E**) Quantification of GFP fluorescence of axon projections in the contralateral cortical layers measured through the region demarked by a discontinuous teal line in insets 2 and 4. Data in **C–E** represent mean ± SEM (shadow) of n = 3–4 controls and n = 3 AnkB440 KO brains. Unpaired *t* test. *p < 0.05, ***p < 0.001, ****p < 0.0001.

The online version of this article includes the following figure supplement(s) for figure 4:

**Source data 1.** Loss of AnkB440 does not alter levels of myelination of CC axons.

cortical neurons and in neurons harboring the p.Y1229H L1CAM variant, which lacks AnkB440-binding activity (*Yang et al., 2019*), at levels indistinguishable from control neurons (*Figure 5—figure supplement 2*, see *Figure 5—source data 1*). These results indicate a selective requirement for AnkB440 for cortical neuron responses to the GC collapse-inducing effect of Sema 3 A.

## The AnkB440-L1CAM complex promotes GC collapse in response to Sema 3A

To confirm that AnkB440 and L1CAM associate at GCs of cultured neurons, we performed a proximity ligation assay (PLA) using antibodies specific for L1CAM and AnkB440, which produces a signal when both proteins are within 40 nm of each other (*Fredriksson et al., 2002*; *Lorenzo and Bennett, 2017*). PLA signal was detected at the GC and throughout the axons in control neurons but was virtually lost in AnkB440 KO neurons and in L1CAM neurons harboring the p.Y1229H variant (*Figure 6A*, see *Figure –source data 1*), consistent with previous findings in vivo and in vitro (*Yang et al., 2019*). We further confirmed the specific interaction of L1CAM with AnkB440 using a cellular assay where co-expression of each GFP-tagged AnkB isoform with L1CAM in HEK293T cells resulted in the effective recruitment of GFP-AnkB440, but not of GFP-AnkB220, from the cytoplasm to the plasma membrane (*Figure 6—figure supplement 1*, see *Figure 6—source data 1*). Notably, treatment of p.Y1229H L1CAM neurons with Sema 3 A resulted in negligible changes in the percent of collapsed GCs in both the axon and collateral axonal branches (*Figure 6B–E*, see *Figure 6—source data 1*). Loss of collateral axon branch GC collapse was also detected by live imaging in AnkB440 KO neurons (*Figure 6D and E*; *Figure 6—video 1*), which could contribute to the aberrant development and connectivity of major axonal tracts. Taken together, these results support a critical role for AnkB440 and the AnkB440-L1CAM complex in Sema 3A-induced GC collapse.

## AnkB440 stabilizes L1CAM and the Sema 3 A receptor neuropilin-1 at the cell surface of GCs

We next sought to identify the mechanisms by which AnkB440 and its interaction with L1CAM promotes GC collapse in response to Sema 3 A. First, we established that AnkB440 loss does not alter L1CAM expression in total lysates from PND1 AnkB440 KO cortex or from AnkB440 KO cortical neuron cultures (*Figure 7A–D*, see *Figure 7—source data 1*). Using a biotinylation assay that labels and captures surface proteins in cortical neuron cultures, we determined that AnkB440 loss markedly reduced surface levels of L1CAM relative to its membrane abundance in control neurons, while surface levels of the AMPA receptor subunit GluR1 remained unchanged (*Figure 7C and D*, see *Figure 7—source data 1*). These findings indicate that AnkB440 is required to maintain normal levels of L1CAM at the cell surface of neurons. L1CAM is not a Sema 3 A receptor (*Castellani et al., 2000*). Instead, L1CAM directly binds the Sema 3 A receptor Nrp1 (*Chen et al., 1998*), which due to its short cytoplasmic domain requires the formation of complexes with transmembrane co-receptors to be stably anchored at the cell surface and propagate repulsive Sema 3 A signals (*Castellani et al., 2000*; *Castellani et al., 2002*). In addition to the L1CAM-Nrp1 complex, Plexin A1 also takes part in the transduction of Sema 3 A signals (*Bechara et al., 2008*). Thus, we next evaluated whether these two receptors participate in AnkB440-L1CAM modulation of Sema 3 A effects. Loss of AnkB440 KO led to a slight increase of Nrp1 levels in total cortical lysates (*Figure 7A and B*) and to a marked reduction of surface-bound Nrp1 in AnkB440 KO cortical neurons, assessed via surface biotinylation, despite its

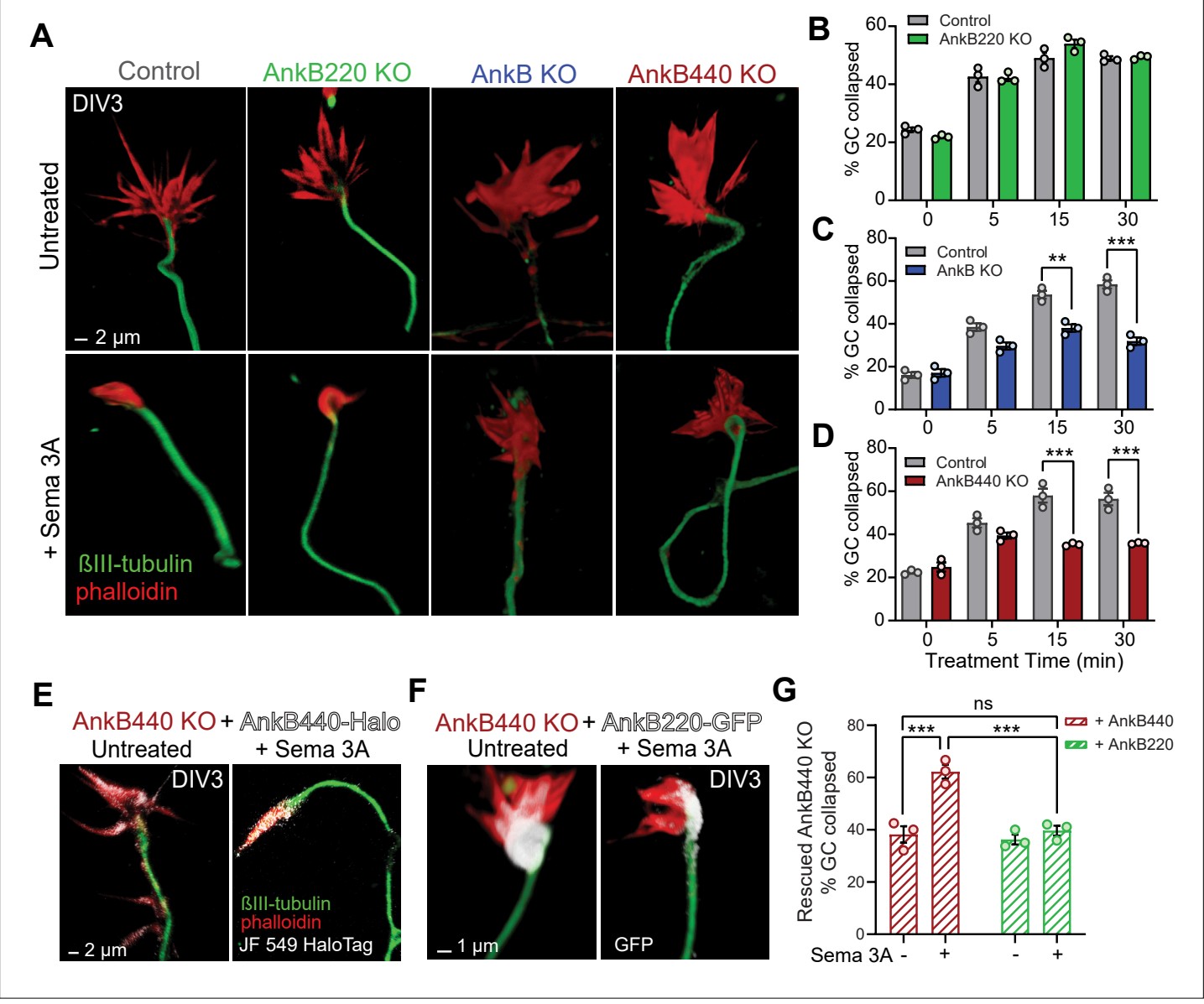

**Figure 5.** AnkB440 is required for Sema 3A-induced GC collapse. (**A**) Images of the distal portion of axons of DIV3 cortical neurons untreated and treated with Sema 3 A and stained with phalloidin and βIII-tubulin. Scale bar, 2 μm. (**B–D**) Percent of GC collapsed before and after Sema 3 A treatment of control, AnkB220 (**B**), total AnkB KO (**C**) and AnkB440 KO (**D**) cortical neurons. Data in B-D represent mean ± SEM collected from an average of n = 120–200 GCs/treatment/genotype per each experiment. Each dot in the graph represents one of three independent experiments. Unpaired *t* test. **p < 0.01, ***p < 0.001. (**E**) Images of DIV3 AnkB440 KO cortical neurons expressing Halo-tagged AnkB440 untreated and treated with Sema 3 A and stained with Janelia Fluor 549 HaloTag ligand to visualize AnkB440. Scale bar, 2 μm. (**F**) Images of DIV3 AnkB440 KO cortical neurons expressing GFP-tagged AnkB220 untreated and treated with Sema 3 A and stained for GFP to visualize AnkB220. Scale bar, 1 μm. (**G**) Percent of GC collapsed before and after Sema 3 A treatment of AnkB440 KO neurons rescued with AnkB440 or AnkB220 cDNAs. Data represent mean ± SEM collected from an average of n = 150 GCs/treatment condition/transfection from three independent experiments. One-way ANOVA with Tukey's post hoc analysis test for multiple comparisons. $^{ns}$p >0.05, ***p < 0.001.

The online version of this article includes the following figure supplement(s) for figure 5:

**Source data 1.** AnkB440 promotes Sema 3A-induced GC collapse.

**Figure supplement 1.** AnkB440 is abundant in axonal growth cones.

**Figure supplement 2.** AnkB440 does not enable Ephrin A5-induced GC collapse.

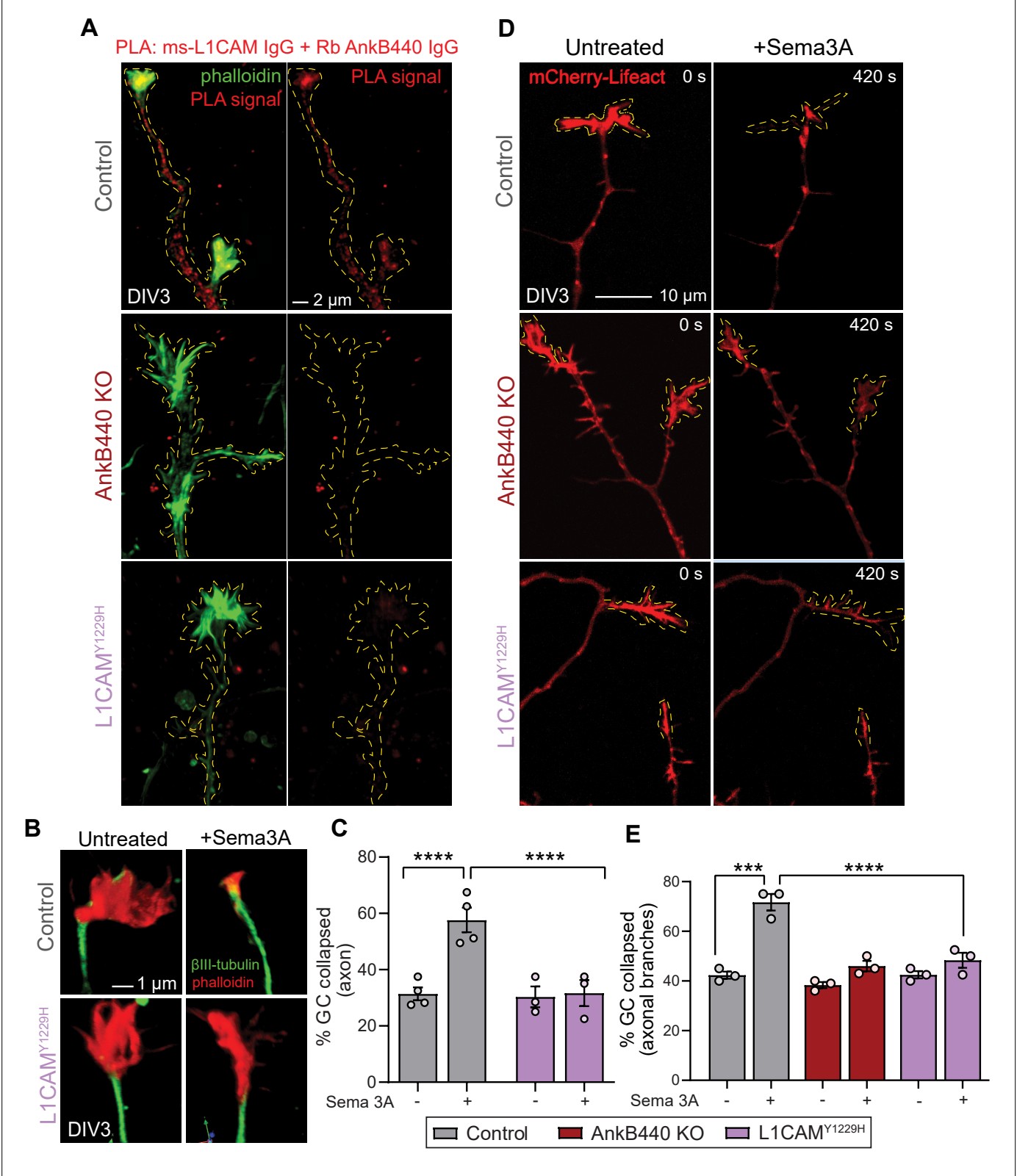

**Figure 6.** The AnkB440-L1CAM complex promotes GC collapse induced by Sema 3 A. (**A**) Images show PLA signal between AnkB440 and L1CAM in axons and GCs of DIV3 cortical neurons from the indicated genotypes. Phalloidin staining was used to identify GCs. Scale bar, 2 µm. (**B**) Images of the distal portion of the main axon untreated and treated with Sema 3 A and stained with phalloidin and βIII-tubulin. Scale bar, 1 µm. (**C**) Percent of axon GCs that collapse before and after Sema 3 A treatment. (**D**) Images from timelapse sequences of DIV3 cortical neuron expressing mCherry-Lifeact

*Figure 6 continued on next page*

*Figure 6 continued*

before and after Sema 3 A treatment. Scale bar, 10 μm. (**E**) Percent of collateral axon branches GCs that collapse before and after Sema 3 A treatment. Data in **C,E** represent mean ± SEM collected from an average of n = 80 GCs/treatment/genotype/experiment. Each dot represents one out of three or four independent experiments. One-way ANOVA with Tukey's post hoc analysis test for multiple comparisons. ***p < 0.001, ****p < 0.001.

The online version of this article includes the following video and figure supplement(s) for figure 6:

**Source data 1.** The AnkB440-L1CAM complex promotes Sema 3A-induced GC collapse.

**Figure supplement 1.** L1CAM recruits AnkB440 to the cell membrane.

**Figure 6—video 1.** The AnkB440-L1CAM complex promotes GC collapse induced by Sema 3 A.
https://elifesciences.org/articles/69815/figures#fig6video1

---

increased total expression (*Figure 7C and D*). In contrast, Plexin A1 levels remained unchanged both in AnkB440 KO brains (*Figure 7A and B*) and at the surface of AnkB440 KO neurons (*Figure 7C and D*). Noticeably, Sema 3 A levels were increased in the cortex of AnkB440 KO PND1 mice (*Figure 7A and B*), which together with the smaller increase in Nrp1 (*Figure 7A–D*) points towards protein upregulation as a likely mechanism to compensate for the observed loss of Sema 3A-mediated signaling responses.

To confirm whether surface levels of L1CAM-Nrp1 complexes were altered in GCs of AnkB440 KO cortical neurons, we selectively labeled surface Nrp1 in non-permeabilized DIV3 neurons with an Nrp1 antibody that recognizes an extracellular epitope. Then, we labeled all L1CAM molecules upon cell permeabilization and visualized surface Nrp1-L1CAM complexes using the PLA assay. PLA signal between surface Nrp1 and L1CAM, indicative of surface Nrp1-L1CAM complexes, was abundant at the GC and in the axon of control neurons, but significantly reduced in AnkB440 KO neurons (*Figure 7E and F*). Consistent with the proposed role of AnkB440 in promoting L1CAM, and consequently, Nrp1 localization at the cell surface, p.Y1229H L1CAM neurons in which the AnkB440-L1CAM association is disrupted, also showed a dramatic reduction in surface L1CAM-Nrp1 PLA signal (*Figure 7E and F*). Both AnkB440 KO and p.Y1229H L1CAM neurons exhibited reduced internalization of surface Nrp1 in response to Sema 3 A (*Figure 7G and H*), consistent with previous reports showing that Nrp1 internalization is required during Sema 3A-induced GC collapse (*Castellani et al., 2004*). Together, these results indicate that AnkB440 promotes the membrane surface localization of the L1CAM-Nrp1 holoreceptor complex, which is required to respond to Sema 3 A repulsive cues.

## AnkB440 does not require βII-Spectrin or binding to microtubules to transduce Sema 3 A signals during GC collapse

AnkB directly binds βII-spectrin (*Davis and Bennett, 1984*; *Davis et al., 2009*), which is widely distributed along axons and required for axonal elongation and organelle transport (*Lorenzo et al., 2019*). βII-spectrin binds F-actin and tetramers of βII-spectrin/αII-spectrin organize a periodic submembrane network of F-actin and associated proteins throughout all axonal domains (*Xu et al., 2013*). Thus, we next investigated whether βII-spectrin is required to propagate Sema 3 A signals to the F-actin network during GC collapse. GCs of cortical neurons harvested from mice lacking βII-spectrin in the brain (βII-spectrin KO) (*Galiano et al., 2012*; *Lorenzo et al., 2019*) collapsed normally in response to Sema 3 A treatment (*Figure 7—figure supplement 1A,B*, see *Figure 7—source data 1*). This result indicates that AnkB440 promotes Sema 3A-induced GC collapse independently of βII-spectrin.

AnkB440 also binds and bundles microtubules through a bipartite microtubule-interaction site located in its NSD (*Chen et al., 2020*). This microtubule-binding activity is required to suppress ectopic axon branching in cultured neurons (*Chen et al., 2020*). The site of microtubule interaction in AnkB440 comprises a module of 15 tandem imperfect 12-aa repeats that includes highly conserved residues in the third (Pro, P), fifth (Ser, S), and ninth positions (Lys, K) (*Chen et al., 2020*). Point mutations in each of these residues in full-length AnkB440 (PSK mutant) is sufficient to impair microtubule-binding activity and cause axon hyperbranching in vitro (*Chen et al., 2020*). To test whether AnkB440-microtubule binding is required for Sema 3A-induced GC collapse, we transfected Halo-tagged cDNA of AnkB440-PSK into AnkB440 KO cortical neurons and treated them with Sema 3 A. Like in control neurons, about 60 % of GCs of neurons expressing AnkB440-PSK collapsed in response to Sema 3 A, indicating that the AnkB440-microtubule interaction is not required for normal transduction of Sema 3 A signals (*Figure 7—figure supplement 1C,D*, see *Figure 7—source data 1*).

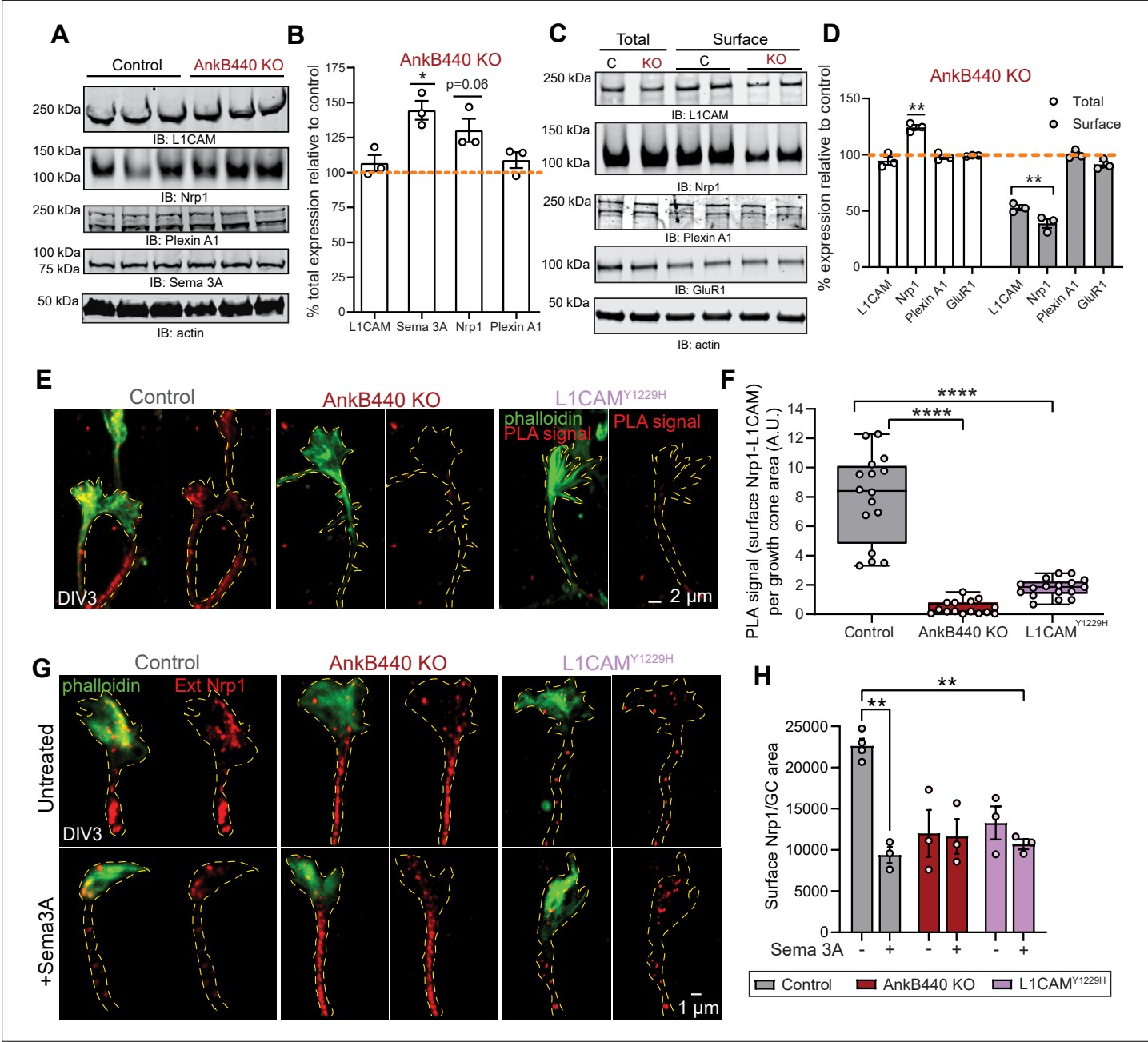

**Figure 7.** AnkB440 stabilizes the Sema 3 A receptor complex L1CAM-Nrp1 at the cell surface of GCs. (**A**) Western blot analysis of the expression of Sema 3 A, L1CAM and the Sema 3 A receptors Nrp1 and Plexin A1 in the cortex of PND1 control and AnkB440 KO mice. (**B**) Quantification of protein levels normalized to actin in cortical lysates from PND1 mice of indicated genotypes relative to the normalized levels of each protein in control brains. Data show mean ± SEM for three biological replicates per genotype for one experiment. Unpaired *t* test. *p < 0.05. (**C**) Western blot analysis of total and surface levels of indicated proteins in DIV3 control and AnkB440 KO cortical neurons. (**D**) Quantification of total and surface protein levels normalized to actin relative to the normalized levels of each protein in control cortical neurons. Data show mean ± SEM for three biological replicates per genotype for one experiment. Unpaired *t* test. **p < 0.01. (**E**) Images show PLA signal between L1CAM and Nrp1 at the cell surface of the axon and GCs of DIV3 cortical neurons from the indicated genotypes. This assay used an Nrp1 antibody that selectively recognizes an extracellular epitope. Phalloidin staining was used to identify GCs. Scale bar, 2 μm. (**F**) Quantification of PLA signal at the GC surface relative to GC area collected from an average n = 15 GCs/genotype. The box and whisker plots represent all data points arranged from minimum to maximum. One-way ANOVA with Tukey's post hoc analysis test for multiple comparisons. ****p < 0.0001. (**G**) Images show Nrp1 localization at the surface of GCs of DIV3 neurons untreated and treated with Sema 3 A, which induces the internalization of surface Nrp1. Scale bar, 1 μm. (**H**) Quantification of surface Nrp1 levels at GCs relative to GC area at the basal state and upon Sema 3A-induced Nrp1 internalization. Data represent mean ± SEM collected from an average of n = 90 GCs/

*Figure 7 continued on next page*

Figure 7 continued

treatment/genotype/experiment. Each dot represents one out of three independent experiments. One-way ANOVA with Tukey's post hoc analysis test for multiple comparisons. **p < 0.01.

The online version of this article includes the following figure supplement(s) for figure 7:

Source data 1. AnkB440 stabilizes the L1CAM-Nrp1 complex at the cell surface of GCs.

Figure supplement 1. AnkB440 does not require βII-spectrin or binding to microtubules to transduce Sema 3 A signaling.

## AnkB440 and its interaction with L1CAM are required to modulate cofilin activity in response to Sema 3A

GC motility and collapse involves fast turnover and reorganization of actin filaments (*Omotade et al., 2017*). F-actin dynamics is regulated by the actin depolymerizing and severing factor (ADF)/cofilin (*Carlier et al., 1997*; *Maciver, 1998*). Cofilin phosphorylation by the Ser/Thr kinase LIM-kinase (LIMK) at the Ser3 site (*Arber et al., 1998*; *Yang et al., 1998*) inactivates cofilin and prevents its F-actin severing activity (*Agnew et al., 1995*; *Moriyama et al., 1996*). Similarly, LIMK phosphorylation of cofilin and the rapid subsequent cofilin activation have been shown to be critical steps in the disassembly of actin filaments during Sema 3A-induced GC collapse (*Aizawa et al., 2001*). Thus, we evaluated whether loss of AnkB440 led to changes in expression of LIMK and cofilin, or in cofilin phosphorylation. Total levels of LIMK and cofilin were not altered in cortical lysates from PND1 AnkB440 brains (*Figure 8A and B*, see *Figure 8—source data 1*). However, AnkB440 loss decreased the ratio of inactive phospho-cofilin (pCofilin$^{S3}$) to total cofilin by 40 % relative to control (*Figure 8A and B*). This increase in active cofilin in AnkB440 KO mice during early brain development might provide a more dynamic actin pool and could underlie the surges in axonal actin patches and emerging filopodia observed in AnkB440 KO cortical neurons.

To determine whether AnkB440 participates in Sema 3A-induced regulation of actin dynamics through the action of cofilin, we examined the effect of Sema 3 A on the localized activation/inactivation of cofilin at GCs through confocal microscopy. As previously observed in cultured dorsal root ganglion neurons (*Aizawa et al., 2001*), levels of pCofilin rose rapidly (above 3-fold) in the GCs of control cortical neurons during the first minute after exposure to Sema 3 A, but underwent a sharp reduction during the next four minutes to 43 % of basal levels (*Figure 8D and E*). This sharp F-actin stabilizing period followed by a fast increase in F-actin depolymerization is thought to reflect the reorganization of the actin cytoskeleton during GC collapse (*Aizawa et al., 2001*). Consistent with this plausible signaling cascade and lesser GC collapse due to a diminished response to Sema 3 A, GCs of both AnkB440 KO and p.Y1229H L1CAM neurons showed a different pattern of cofilin phosphorylation. First, basal pCofilin signal per GC area was roughly 40% and 30% lower in AnkB440 KO and p.Y1229H L1CAM neurons, respectively, relative to control (*Figure 8D and E*), which is consistent with lower levels of pCofilin in AnkB440 brains (*Figure 8A and C*). Second, in contrast to the higher than three-fold increase in control neurons, the rise in pCofilin during the first minute of Sema 3 A treatment was below two-fold in both AnkB440 KO and p.Y1229H L1CAM GCs, which represented only 30–40% of control levels. Lastly, in AnkB440 KO and p.Y1229H L1CAM GCs the reduction of pCofilin 5 min past Sema 3 A treatment was approximately only 20 % lower than its peak at one minute and remained around 50 % above basal levels (*Figure 8D and E*). These aberrant patterns of cofilin regulation indicate that AnkB440 and its association with L1CAM promote steps of the Sema 3 A signal transduction pathways upstream of changes in cofilin activation and F-actin disassembly during GC collapse.

## Autism-linked *ANK2* variants affect the transduction of Sema 3A repulsive cues

The results above support a mechanism wherein AnkB440 is required to modulate the initiation, growth, and establishment of axon collateral branches and to properly respond to Sema 3 A repulsive cues that modulate GC collapse, axon guidance and targeting, and pruning. Consequently, AnkB440 deficiencies can result in aberrant structural and functional axon connectivity, which in turn may contribute to the pathogenicity of *ANK2* variants in ASD. AnkB440$^{R2589fs}$ mice, which models the de novo p.(R2608fs) frameshift variant in exon 37 of *ANK2* found in an individual diagnosed with ASD, exhibit ectopic axonal connections assessed by brain DTI and axonal hyperbranching in vitro (*Yang*

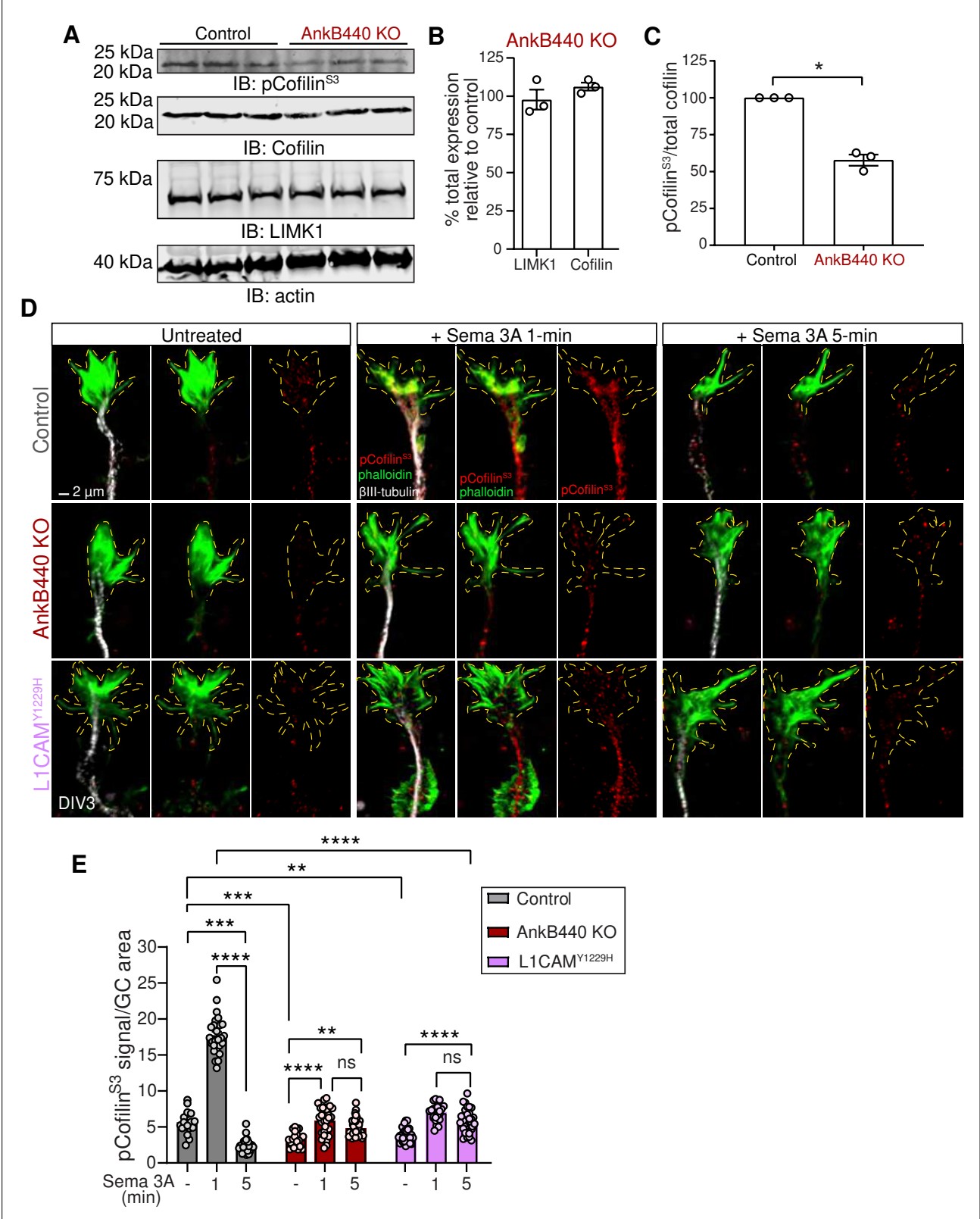

**Figure 8.** AnkB440 and its interaction with L1CAM are required for F-actin disassembly during GC collapse upon Sema 3 A treatment. (**A**) Western blot analysis of the expression of cofilin, phospho-cofilin (Ser3) (pCofilin$^{Ser3}$) and LIMK1 in the cortex of PND1 control and AnkB440 KO mice. Actin is a loading control. (**B**) Quantification of total levels of cofilin and LIMK1 normalized to actin in cortical lysates from PND1 AnkB440 KO mice relative to the normalized levels of each protein in control brains. (**C**) Quantification of levels of pCofilin$^{Ser3}$ relative to total cofilin in cortical lysates from PND1 mice.

*Figure 8 continued on next page*

*Figure 8 continued*

Data in **B** and **C** represent mean ± SEM for three biological replicates per genotype. Unpaired *t* test. *p < 0.05. (**D**) Images of the distal portion of the main axon of DIV3 cortical neurons untreated and treated with Sema 3 A for 1 and 5 minutes and stained with phalloidin, βIII-tubulin, and pCofilin$^{Ser3}$. Dotted lines indicate GCs. Scale bar, 2 µm. (**E**) Quantification of pCofilin$^{S3}$ signal at GCs relative to GC area at the basal state and upon Sema 3 A treatment for 1 and 5 min. Data represent mean ± SEM from an average of n = 25–40 GCs/treatment/genotype collected from three independent experiment. One-way ANOVA with Tukey's post hoc analysis test for multiple comparisons. $^{ns}$p >0.05, **p < 0.01, ***p < 0.001, ****p < 0.0001.

The online version of this article includes the following figure supplement(s) for figure 8:

**Source data 1.** AnkB440 interaction with L1CAM promotes F-actin disassembly during Sema 3A-induced GC collapse.

*et al., 2019*). Instead of full-length AnkB440, AnkB440$^{R2589fs}$ mice express a 290 kDa truncated product that lacks portions of the NSD unique to AnkB440 plus the entire death and C-terminal regulatory domains (*Yang et al., 2019*; *Figure 9A and B* asterisk, see *Figure 9—source data 1*). Given that this truncated protein fails to associate with L1CAM in vivo (*Yang et al., 2019*), we tested whether its expression could also disrupt Sema 3A-induced GC collapse. Like AnkB440 KO and p.Y1229H L1CAM neurons, AnkB440$^{R2589fs}$ neurons had diminished GC collapse response to Sema 3 A (*Figure 9C and D*). Thus, in addition to altered microtubule stability, impaired responses to Sema 3 A during GC collapse likely contribute to the axonal branching and connectivity deficits observed in AnkB440$^{R2589fs}$ brains and neurons.

Over 70 *ANK2* variants have been identified in individuals diagnosed with ASD (*De Rubeis et al., 2014*; *Iossifov et al., 2014*; *Iossifov et al., 2015*). ASD-linked *ANK2* variants target the NSD or domains shared by both AnkB440 and AnkB220 isoforms. We previously reported that de novo ASD variants p.(P1843S) and p.(E3429V) in the NSD of AnkB440 (*Figure 9A*) failed to rescue axon hyper-branching of AnkB440 KO cortical neurons (*Yang et al., 2019*). Therefore, we evaluated whether these AnkB440-specific variants could restore GC responses to Sema 3 A of AnkB440 KO neurons. We also tested expression of Halo-tagged AnkB440 bearing the de novo variant p.(R1145Q) ASD variant in exon 30, which encodes a portion of the ZU5$^C$ domain common to both AnkB220 and AnkB440 (*Figure 9A*). AnkB440 KO cortical neurons expressing these ASD-linked AnkB440 variants failed to rescue GC collapse in response to Sema 3 A (*Figure 9E and F*). These variants do not appear to affect protein stability, given that they expressed normal levels of full-length AnkB440 protein when transfected in HEK293T cells, which lack endogenous AnkB440 expression (*Figure 9G*, see *Figure 9—source data 1*). Instead, they likely disrupt AnkB440 distribution, as in the case of the p.(P1843S) AnkB variant, which exhibited reduced Halo-AnkB440 localization to the GCs (*Figure 9E and F*), or might disrupt interaction with specific partners. Together, these results suggest that AnkB440 variants can lead to loss-of-function effects that may affect structural axon connectivity and contribute to pathogenicity in ASD.

## Discussion

Variants in ankyrin genes have emerged as risk factors in multiple neurodevelopmental and psychiatric disorders. For example, several GWAS and other genetic studies have found both rare and common variants in *ANK3*, which encodes ankyrin-G (AnkG), associated with bipolar disorder (*Baum et al., 2008*; *Ferreira et al., 2008*) and schizophrenia (*Cruz et al., 2009*; *Schizophrenia Psychiatric Genome-Wide Association Study (GWAS) Consortium, 2011*). Similarly, variants in *ANK2* have been identified in individuals with ASD and intellectual disability (*De Rubeis et al., 2014*; *Iossifov et al., 2014*; *Iossifov et al., 2015*) and it is ranked as a top high confidence ASD gene with one of the highest mutability scores (*Ruzzo et al., 2019*). Despite their structural similarities and degree of sequence conservation, ankyrins diverge in cell type expression, subcellular localization, and protein partners in the brain (*Lorenzo, 2020*). For instance, AnkG preferentially localizes to the axon initial segment (AIS), where it acts as the master regulator of AIS organization, while AnkB is widely distributed throughout the axon (*Lorenzo et al., 2014*; *Yang et al., 2019*; *Lorenzo, 2020*). Ankyrins achieve a second order of functional specialization through alternative splicing, which yields giant isoforms with unique inserted sequences in both AnkG and AnkB in neurons (*Bennett and Lorenzo, 2016*; *Lorenzo, 2020*). ASD-linked AnkB variants fall both within the inserted region unique to AnkB440 and in domains shared by AnkB440 and AnkB220 isoforms.

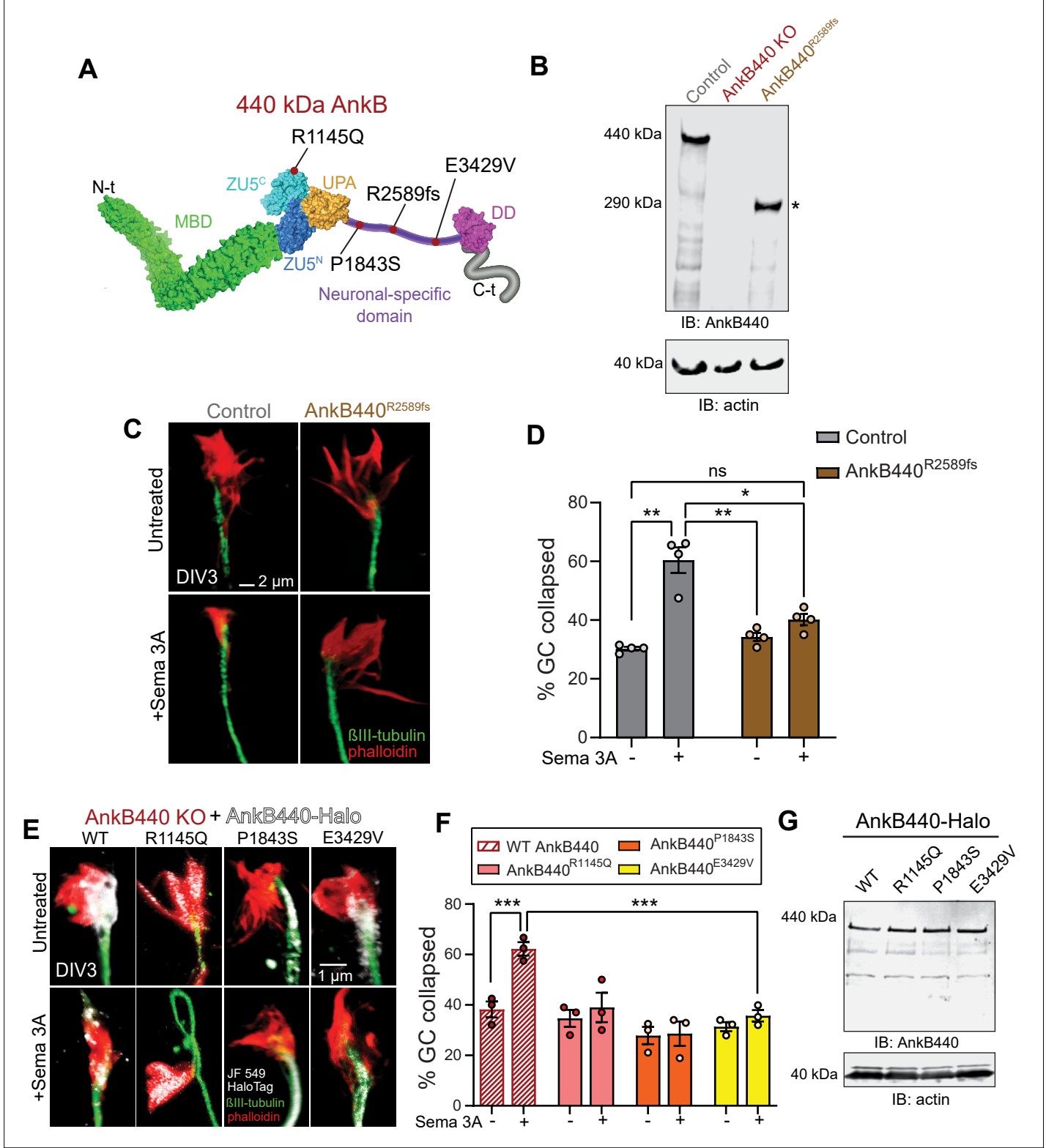

**Figure 9.** ASD-linked AnkB440 variants affect the transduction of Sema 3 A cues during GC collapse. (**A**) Red dots indicate the position within AnkB440 functional domains of the ASD-linked *ANK2* variants evaluated. (**B**) Western blot analysis of expression of AnkB440 in cortical lysates of PND1 control mice of the indicated genotypes assessed with an AnkB440-specific antibody. Brains of AnkB440R2589fs mice express a truncated 290 kDa AnkB440 fragment (asterisk). (**C**) Images of the distal portion of the axon of control and AnkB440R2589fs DIV3 neurons untreated and treated with Sema 3 A and stained with phalloidin and βIII-tubulin. Scale bar, 2 μm. (**D**) Percent of axon GCs that collapse before and after Sema 3 A treatment. Data represent

*Figure 9 continued on next page*

*Figure 9 continued*

mean ± SEM collected from an average of n = 130 GCs/treatment/genotype. Each dot represents one out of four independent experiments. One-way ANOVA with Tukey's post hoc analysis test for multiple comparisons. $^{ns}p > 0.05$, $*p < 0.05$, $**p < 0.01$. (**E**) Images of the distal portion of the axon of DIV3 AnkB440 KO cortical neurons rescued with the indicated AnkB440-Halo plasmids untreated and treated with Sema 3 A. Scale bar, 1 μm. (**F**) Percent of axon GCs that collapse before and after Sema 3 A treatment. Data represent mean ± SEM collected from an average of n = 70 GCs/treatment/genotype. Each dot represents one out of three independent experiments. One-way ANOVA with Tukey's post hoc analysis test for multiple comparisons. $***p < 0.001$. (**G**) Western blot analysis of expression of AnkB440-Halo plasmids in HEK293T cells. Data is representative of three independent experiments.

The online version of this article includes the following figure supplement(s) for figure 9:

**Source data 1.** ASD-linked AnkB440 variants affect the transduction of Sema 3 A cues during GC collapse.

The present study sheds new light into the isoform-specific functions of AnkB in modulating axonal architecture, targeting, and refinement in the developing brain and the potential contribution of AnkB deficits to ASD pathology. We previously showed that expression of the truncated product of the frameshift mutation p.P2589fs that models the human de novo p.(R2608fs) ASD variant in AnkB440 causes stochastic increases in structural cortical connectivity in mouse brains and exuberant collateral axon branching in cortical neuron cultures (*Yang et al., 2019*). We confirmed the development of axon hyperbranching in cultured neurons selectively lacking AnkB440 (*Chen et al., 2020*), but not in neurons from AnkB220 KO mice. Using Golgi staining, we determined that AnkB440 KO, but not AnkB220 KO, cortical neurons grow more collateral branches in the proximal axon, consistent with the in vitro results. We also found volumetric increases of multiple commissural and other axon tracts in AnkB440 KO mice and deficits in topographic order of midline CC axons arising from the somato-sensory cortex and the targeting of their contralateral projections. These changes in cortical structural connectivity likely result from combined defects in axon branching initiation, fasciculation, guidance, and pruning of ectopic projections during development. These findings support critical and specialized roles of AnkB440 in modulating both axon collateral branch formation and refinement.

Previous results implicated AnkB440 in suppressing collateral branch formation via the stabilization of microtubule bundles near the plasma membrane (*Yang et al., 2019*; *Chen et al., 2020*). Loss of AnkB440 or expression of the 290 kDa truncated AnkB440 product promotes microtubule unbundling, which facilitates microtubule invasion of the nascent filopodia, a necessary step in the commitment to forming a new branch (*Yu et al., 1994*). The establishment of transient actin nucleation sites marks the point of formation and precedes the emergence of the precursor filopodia (*Gallo, 2011*). In this study, we show that selective loss of AnkB440 KO in cultured cortical neurons results in a larger number of actin patches relative to control and AnkB220 KO neurons, which correlated with higher number of axonal filopodia and collateral branches. Thus, AnkB440 modulates collateral branch formation through a combined mechanism that suppresses the stochastic formation of collateral filopodia by lowering the density of F-actin-rich branch initiation points, as well as the invasion of microtubules into the maturing filopodia. Although F-actin patches serving as precursor to filopodia have been observed in vitro (*Loudon et al., 2006*; *Spillane et al., 2011*) and in vivo (*Spillane et al., 2011*; *Hand et al., 2015*), little is known about the factors that regulate their formation and dynamics (*Armijo-Weingart and Gallo, 2017*). How AnkB440 modulate actin patches is not clear. One possibility is that AnkB440 associates with actin regulators that suppress the availability of actin monomers and/or of branched actin required for actin nucleation, and the formation of focal F-actin patches (*Spillane et al., 2011*). Interestingly, we found that AnkB440 loss increases active levels of cofilin, a positive regulator of actin turnover and dynamics, neurite outgrowth, and axon branching (*Flynn et al., 2012*; *Tedeschi et al., 2019*). Alternatively, AnkB440 may associate with actin through their common partner βII-spectrin, although loss of βII-spectrin does not lead to axonal hyperbranching in vitro and may not affect the number or dynamics of actin patches (*Lorenzo et al., 2019*). The crosstalk between actin and microtubule networks has also been proposed to orchestrate the emergence of branches (*Dent and Kalil, 2001*; *Pacheco and Gallo, 2016*), and splaying of microtubule bundles have been observed to correlate with F-actin accumulation at branch points (*Dent and Kalil, 2001*). Thus, it is possible that microtubule unbundling resulting from AnkB440 deficits promotes the seeding and growth of F-actin patches at sites of axon branch formation. While loss of AnkB220 does not affect axonal branching in vitro and in mouse brains, its expression in AnkB440 null neurons may be necessary to sustain hyperbranching. This is possibly due to the role of AnkB220 in axonal organelle and vesicle transport

and growth, which are not affected by exclusive loss of AnkB440. It will be important to investigate how AnkB440 modulates actin dynamics and its effects on other actin-dependent processes during neuronal and brain development.

While cell-autonomous factors that modulate axonal branching in vitro, such as the formation of actin patches, correlate with the number of axon branches (*Gallo, 2011*), live imaging of actin dynamics during in vivo axonal development found that this correlation does not hold in projection neurons in the mouse cortex (*Hand et al., 2015*). Instead, the number of collateral axon branches along the axon is determined by the cortical layer they transverse, indicating that extrinsic factors that modulate branch formation and pruning may be involved in determining the pattern of axon innervation (*Hand et al., 2015*). Several reports support the activity of Sema 3 A as an extrinsic repellent cue that inhibits axon branching in vitro (*Dent et al., 2004*) and prunes cortical axons and collateral axon branches in vivo (*Polleux et al., 1998*; *Bagri et al., 2003*). Nrp1-mediated Sema3A signaling also promotes the topographic order of axons from different cortical areas within the CC midline and their homotopic contralateral targeting (*Zhou et al., 2013*; *Martín-Fernández et al., 2021*). We show that AnkB440 is required to transduce repellent Sema 3 A signals in vitro to facilitate the collapse of GCs from the primary axon and collateral branches, which offers a plausible cellular mechanism underlying the ectopic axon connectivity, deficits in developmental axon refinement, and the axon hyperbranching observed in AnkB440-deficient mouse brains. Impaired transduction of Sema 3 A likely also contributes to the disrupted topographic organization of S1-derived callosal axons at the CC midline and their aberrant contralateral targeting. The observed CC defects do not involved oligodendrocytes, which do not express measurable amounts of AnkB440. However, neuronal loss of AnkB440 resulted in loose myelin wraps in the CC, which are possibly indicative of myelin decompaction and warrants further investigation.

AnkB440-mediated transduction of Sema 3 A during GC collapse involves AnkB440 binding to L1CAM, which in turn stabilizes the Sema 3 A holoreceptor complex L1CAM-Nrp1 at the cell surface (*Castellani et al., 2000*; *Castellani et al., 2002*; *Castellani et al., 2004*). Interestingly, GCs of neurons lacking the F-actin and AnkB partner βII-spectrin respond normally to Sema 3 A, which indicates that βII-spectrin is not required for Sema 3A-induced GC collapse. However, given βII-spectrin's role in the development and wiring of axons in mouse brains (*Galiano et al., 2012*; *Lorenzo et al., 2019*) and the recent identification that pathogenic variants in the βII-spectrin encoding gene *SPTBN1* cause a neurodevelopmental syndrome associated with deficits in cortical connectivity (*Cousin et al., 2021*), we cannot rule out its involvement in axonal guidance through alternative mechanisms. Interestingly, although AnkB440 binding to microtubules suppresses branch initiation, loss of this interaction does not affect the response to Sema 3 A during GC collapse. Instead, transduction of Sema 3 A signaling via the AnkB440-L1CAM-Nrp1 complex modulates F-actin dynamics through LIMK phosphorylation of cofilin (*Figure 10*).

Our structure-function studies in AnkB440 KO cortical neurons found that de novo ASD variants p.(R1145Q), p.(P1843S), p.R2589fs, and p.(E3429V) in AnkB440 fail to restore cellular responses to Sema 3 A during GC collapse. Interestingly, expression of the p.R2589fs variants caused axon hyperbranching in vitro and ectopic axon connectivity in vivo, and the remaining variants did not rescue normal collateral branching in AnkB440 KO neurons in vitro (*Yang et al., 2019*). The 290 KDa AnkB440 product of p.R2589fs is expressed at slightly reduced levels compared to full-length AnkB440 and shows reduced binding to L1CAM in vivo (assessed through PLA) despite preserving the L1CAM binding region in the membrane binding domain (MBD) (*Yang et al., 2019*). This diminished AnkB440$^{R2589fs}$-L1CAM association may result from the lower expression of the truncated AnkB440 protein (*Yang et al., 2019*), although it is also possible that this truncation may induce changes in the conformation or self-regulation (*Chen et al., 2017*) of truncated AnkB440 that might interfere with its association with L1CAM. The mechanisms by which the other ASD AnkB440 variants evaluated disrupt AnkB440 function are less clear. The p.(P1843S) might alter AnkB440 distribution. This variant also falls within the microtubule binding region of AnkB440 (*Chen et al., 2020*) and may affect microtubule binding, which if true, may explain the axon hyperbranching phenotype, but it is not clear how it interferes with GC collapse. Both ASD variants p.(R1145Q) and p.(E3429V) in AnkB440 reside outside of the MBD and the microtubule-binding regions. p.(R1145Q) is located in the ZU5$^C$ domain shared by AnkB220 and AnkB440 and it may affect AnkB's ability to bind PI3P lipids, given its proximity to the PI3P lipids binding region (*Lorenzo et al., 2014*). However, while AnkB220 requires binding to PI3P

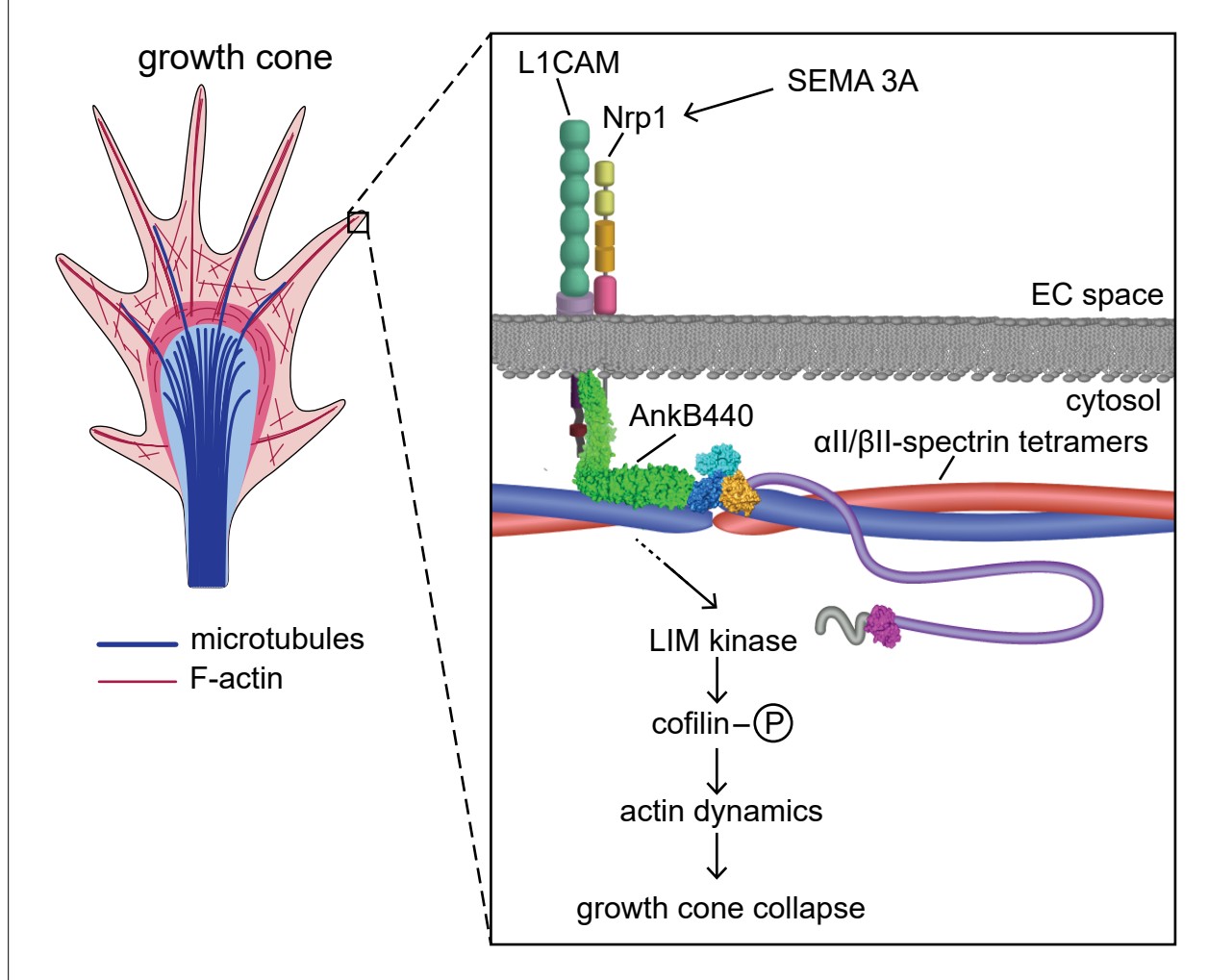

**Figure 10.** Proposed mechanism of AnkB440-mediated growth cone collapse. AnkB440 binds the cytoplasmic domain of L1CAM to stabilize the L1CAM-Nrp1 holoreceptor complex at the cell surface. Transduction of Sema 3 A signals via the AnkB440-L1CAM-Nrp1 complex modulates F-actin dynamics through LIMK phosphorylation of cofilin to facilitate local F-actin disassembly and the collapse of GCs from the axon and collateral branches independently of βII-spectrin, or of AnkB440 interaction with microtubules. **Source data.** Source data contain the original files of the full raw unedited gels or blots and figures with the uncropped gels or blots with the relevant bands clearly labelled. Source data files also include numerical data that are represented as graphs in the figures.

lipids to associate with intracellular membranes and enable axonal transport (*Lorenzo et al., 2014*), AnkB440 does not modulate axonal transport. In future experiments it would be important to determine whether AnkB440 binds PI3P lipids and the significance of AnkB440-PI3P lipid binding activity for axonal development and axonal connectivity in the developing brain.

Besides the significance of *ANK2* in ASD, neuronal pathways involving the AnkB440-L1CAM complex are also relevant to other neurological diseases. For example, a few hundred variants in L1CAM have been described in individuals with CC hypoplasia, retardation, adducted thumbs, apasticity and hydrocephalus (CRASH) syndrome (*Rosenthal et al., 1992*; *Jouet et al., 1994*; *Weller and Gärtner, 2001*; *Vos et al., 2010*). Pathological L1CAM variants can affect both its extra- and intracellular domains and disrupt binding to molecular partners including AnkB and Nrp1 (*Schäfer and Altevogt, 2010*). Studies in mouse models that constitutively lack L1CAM have reported CC hypoplasia, cerebellar and other brain malformations, and axon guidance defects in the corticospinal tract (*Dahme et al., 1997*; *Cohen et al., 1998*; *Fransen et al., 1998*). Interestingly, neurons differentiated from human embryonic stem (ES) cells in which expression of endogenous L1CAM was knockdown through homologous recombination showed reduced axonal length and deficient axonal branching

eLife Research article

Cell Biology | Neuroscience

relative to matching control neurons (*Patzke et al., 2016*). These results are in contrast with the cellular phenotypes we observe in AnkB440 KO neurons, even though ES cell-derived L1CAM KO neurons showed noticeable downregulation of AnkB, which appear to have been largely driven by significant loss of AnkB440. We found that loss of AnkB440 reduces L1CAM abundance at the cell surface but does not significantly change total levels of L1CAM in the cortex or in cortical neuron cultures. L1CAM expression is also normal in brains of AnkB440$^{R2589fs}$ mice (*Yang et al., 2019*). While it is plausible that normal AnkB expression requires L1CAM, and not the reverse, it would be important to confirm whether these changes in protein expression are specific to human ES cell-derived neurons. The phenotypic differences between AnkB440- and L1CAM-deficient mice point to additional, independent pathways involving AnkB440 and L1CAM, including the functional relationship of L1CAM with other ankyrins, which collectively underscore the importance of these proteins in neuronal structure and signaling. For instance, a recent report implicates an L1CAM-AnkG association at the AIS of neocortical pyramidal neurons in their innervation by GABAergic chandelier cells (*Tai et al., 2019*). AnkB440 is widely distributed through all axonal domains, including the AIS. Further work will be needed to determine whether AnkB440-L1CAM complexes localize at the AIS and if they contribute to this specific type of pyramidal neuron innervation.

In summary, our findings unveil a critical role of AnkB440 and its binding to L1CAM in promoting the clustering of L1CAM-Nrp1 complexes at the cell surface of GCs of cortical neurons. Accordingly, we show that the AnkB440-L1CAM-Nrp1 complex enables the chemorepellent action of Sema 3 A to induce GC collapse in axons and collateral branches, thereby providing novel insight into the mechanisms of axonal guidance, topographic ordering of CC axons, contralateral callosal axon targeting, and collateral branch pruning. As we show, *ANK2* variants cause disruption of this signaling axis, which might lead to cortical miswiring and contribute to the neuropathology of ASD.

# Materials and methods

## Key resources table

| Reagent type (species) or resource | Designation | Source or reference | Identifiers | Additional information |
|---|---|---|---|---|
| Genetic reagent (*M. musculus*) | *Ank2*$^{440flox}$ | PMID:31285321 | MGI ID: 6784039. Pending RRID assignment | Dr. Vann Bennett, Duke University |
| Genetic reagent (*M. musculus*) | *Ank2*$^{Exon22-Neo}$ | PMID:25533844 Jackson Laboratory | RRID:IMSR_JAX:027916, Stock No: 027916 | Dr. Vann Bennett, Duke University |
| Genetic reagent (*M. musculus*) | *Ank2*$^{220flox}$ | This paper | | See methods- Generation of AnkB220 knockout mice |
| Genetic reagent (*M. musculus*) | *Ank2*$^{R2589fs}$ | PMID:31285321 | | Dr. Vann Bennett, Duke University |
| Genetic reagent (*M. musculus*) | *L1CAM*$^{Y1229H}$ | PMID:31285321 | | Dr. Vann Bennett, Duke University |
| Genetic reagent (*M. musculus*) | *Sptbn1*$^{flox}$ | PMID:31209033 Jackson Laboratory | RRID:IMSR_JAX:020288, Stock No: 020288 | Dr. Matthew N. Rasband, Baylor College of Medicine |
| Genetic reagent (*M. musculus*) | *Nestin-Cre* | PMID:31209033 Jackson Laboratory | RRID:IMSR_JAX:003771, Stock No: 003771 | Dr. Rudiger Klein, Max-Planck Institute of Neurobiology |
| Genetic reagent (*M. musculus*) | *Thy1-GFP line M* | PMID:11086982 Jackson Laboratory | RRID:IMSR_JAX:007788, Stock No: 007788 | Dr. Joshua R Sanes, Harvard University |
| Cell line (*Homo sapiens*) | HEK 293T/17 | American Type Culture Collection | Cat# CRL-11268, RRID:CVCL_1926 | |
| Antibody | anti-pan ankyrin-B (Rabbit polyclonal) | PMID:31209033, 31285321 | | IF (1:500), WB (1:5000) |
| Antibody | anti-440 kDa ankyrin-B (Rabbit polyclonal) | PMID:31285321 | | IF (1:500), WB (1:5000) |
| Antibody | anti-Sema 3 A (Rabbit polyclonal) | Abcam | Cat# ab23393, RRID:AB_447408 | WB (1:500) |

*Continued on next page*

*Continued*

| Reagent type (species) or resource | Designation | Source or reference | Identifiers | Additional information |
|---|---|---|---|---|
| Antibody | anti-Nrp1 (Rabbit polyclonal) | Abcam | Cat# ab25998, RRID:AB_448950 | WB (1:1000) |
| Antibody | anti-Mbp (Rat monoclonal) | Abcam | Cat# ab7349, RRID:AB_305869 | IF (1:100), WB (1:500) |
| Antibody | anti-L1CAM (mouse monoclonal) | Abcam | Cat# ab24345, RRID:AB_448025 | WB (1:200) |
| Antibody | anti-Satb2 (Mouse monoclonal) | Abcam | Cat# ab51502, RRID:AB_882455 | IF (1:200) |
| Antibody | anti-Tbr1 (Rabbit polyclonal) | Abcam | Cat# ab31940, RRID:AB_2200219 | IF (1:200) |
| Antibody | anti-Ctip2 (Rat monoclonal) | Abcam | Cat# ab18465, RRID:AB_2064130 | IF (1:500) |
| Antibody | anti-Plexin A1 extracellular (Rabbit polyclonal) | Alomone | Cat# APR-081, RRID:AB_2756765 | WB (1:200) |
| Antibody | anti-Nrp1 extracellular (Rabbit polyclonal) | Alomone | Cat# ANR-063, RRID:AB_2756695 | IF(1:50) |
| Antibody | anti-mGluR1, clone N355/1 (Mouse polyclonal) | Antibodies Incorporated | Cat# 75–327, RRID:AB_2315840 | WB (1:500) |
| Antibody | anti-phospho-Cofilin (Ser3) (Rabbit polyclonal) | Cell Signaling Technology | Cat# 3311, RRID:AB_330238 | WB (1:1000) |
| Antibody | anti-LIMK1 (Rabbit polyclonal) | Cell Signaling Technology | Cat# 3842, RRID:AB_2281332 | WB (1:1000) |
| Antibody | anti-MAG (Rabbit monoclonal) | Cell Signaling Technology | Cat# 9043, RRID:AB_2665480 | WB (1:1000) |
| Antibody | anti-MOG (Rabbit monoclonal) | Cell Signaling Technology | Cat# 96457, RRID:AB_2800265 | WB (1:1000) |
| Antibody | anti-GFAP (mouse monoclonal) | Cell Signaling Technology | Cat# 3670, RRID:AB_561049 | WB (1:1000) |
| Antibody | anti-cofilin (Mouse monoclonal) | Proteintech | Cat# 66057–1-Ig, RRID:AB_11043339 | WB (1:1000) |
| Antibody | anti-Olig2 (Rabbit polyclonal) | Proteintech | Cat# 13999–1-AP, RRID:AB_2157541 | IF (1:200) |
| Antibody | Anti-pan neurofilament (mouse monoclonal) | Biolegend | Cat# 837904, RRID:AB_2566782 | IF (1:200) |
| Antibody | anti-actin (mouse monoclonal) | Millipore-Sigma | Cat# MAB1501, RRID:AB_2223041 | WB (1:2000) |
| Antibody | anti-CNPase (mouse monoclonal) | Millipore-Sigma | Cat# C5922, RRID:AB_476854 | WB (1:500) |
| Antibody | anti-NeuN (guinea pig polyclonal) | Millipore-Sigma | Cat# ABN90P, RRID:AB_2341095 | IF (1:1000) |
| Antibody | anti-L1CAM (rat monoclonal) | Millipore-Sigma | Cat# MAB5272, RRID:AB_2133200 | IF (1:200) |
| Antibody | anti- βIII-tubulin (mouse monoclonal) | Millipore-Sigma | Cat# MAB1637, RRID:AB_2341095 | IF (1:200) |
| Antibody | anti-myelin PLP (rabbit polyclonal) | Novus Biologicals | Cat# NBP1-87781, RRID:AB_11026674 | WB (1:1000) |
| Antibody | anti-pan tubulin (sheep polyclonal) | Cytoskeleton | Cat# ATN02, RRID:AB_10709401 | IF (1:200) |

*Continued on next page*

*Continued*

| Reagent type (species) or resource | Designation | Source or reference | Identifiers | Additional information |
|---|---|---|---|---|
| Antibody | anti-GFP (chicken polyclonal) | Antibodies Incorporated | Cat# GFP-1020, RRID:AB_10000240 | IF (1:1000) |
| Recombinant DNA reagent | pBa-AnkB440-Halo | PMID:31285321 | | pBa-Halo backbone |
| Recombinant DNA reagent | pBa-AnkB440$^{P1843S}$-Halo pBa-AnkB440$^{E3429V}$-Halo; | PMID:31285321 | | pBa-Halo backbone |
| Recombinant DNA reagent | pBa-AnkB440$^{R1145Q}$-Halo | This paper | | pBa-Halo backbone. See methods- plasmids |
| Recombinant DNA reagent | pBa-AnkB440-PSK-Halo | PMID:32640013 | | pBa-Halo backbone |
| Recombinant DNA reagent | pCAG-AnkB220-GFP | This paper | | pCAG-eGFP-N1 backbone. See methods- plasmids |
| Recombinant DNA reagent | pBa-AnkB440-GFP | This paper | | pBa-Halo backbone. See methods- plasmids |
| Commercial assay or kit | GeneArt Site-Directed Mutagenesis System | ThermoFisher Scientific | Cat. #: A13282 | |
| Commercial assay or kit | In-Fusion Snap Assembly Starter Bundle | Takara | Cat. #: 638,945 | |
| Commercial assay or kit | DUOLINK PLA in situ detection kit-red | Millipore-Sigma | Cat. #: DUO92008-100RXN | |
| Commercial assay or kit | Duolink In Situ PLA Probe Anti-Mouse PLUS | Millipore-Sigma | Cat. #: DUO92001-100RXN | |
| Commercial assay or kit | Duolink In Situ PLA Probe Anti-Rabbit MINUS | Millipore-Sigma | Cat. #: DUO92005-100RXN | |
| chemical compound, drug | Alexa Fluor- 488,–568, –633 Phalloidin | ThermoFisher Scientific | Cat. #: A12379, A12380, A22284 | IF (1:100) |
| Peptide, recombinant protein | Semaphorin 3 A | Peprotech | Cat. #: 150–17 H | 250 ng/ml |
| Peptide, recombinant protein | Ephrin A5 | R&D Systems | Cat. #: 374-EA | 1 µg/ml |

## Mouse Lines and Animal Care

Experiments were performed in accordance with the guidelines for animal care of the Institutional Animal Care and Use Committee at the University of North Carolina at Chapel Hill. The total AnkB knockout mice (total AnkB KO), the conditional AnkB440 (*Ank2$^{440flox}$*) mouse line, the knock-in AnkB440$^{R2589fs}$ (*Ank2$^{R2589fs}$*) mouse line that models a human ASD-linked *ANK2* variant, and the mice carrying the Y1229H mutation in L1CAM (L1CAM$^{Y1229H}$) were a gift from Dr. Vann Bennett at Duke University and have been previously reported (*Scotland et al., 1998*; *Lorenzo et al., 2014*; *Yang et al., 2019*; *Chen et al., 2020*). βII-spectrin floxed mice (*Sptbn1$^{flox/flox}$*, a gift from Dr. Mathew Rasband) have been previously reported (*Galiano et al., 2012*; *Lorenzo et al., 2019*). AnkB440 KO and AnkB220 KO (described below) mice respectively lacking AnkB440 and AnkB220 in neural progenitors were generated by crossing *Ank2$^{440flox/440flox}$* or *Ank2$^{220flox/220flox}$* animals to the Nestin-Cre line [B6.Cg-Tg(Nes-cre)1Kln/J, stock number 003771] from The Jackson Laboratory. A similar breeding strategy was used to generate mice with loss of βII-spectrin in neural progenitor. Control and AnkB440 KO mice expressing GFP in a subset of neurons were generated by respectively crossing *Ank2$^{440flox/440flox}$* and *Ank2$^{440flox/440flox}$*::Nestin-Cre mice to the Thy1-GFP-M mouse line [Tg(Thy1-EGFP)MJrs/J, stock number 007788] from The Jackson Laboratory (*Feng et al., 2000*). All mice were housed at 22°C ± 2°C on a 12-hour-light/12-hour-dark cycle and fed ad libitum regular chow and water.

## Generation of Conditional AnkB220 Knockout Mice

Mice carrying a floxed allele that selectively targets the AnkB220 isoform (*Ank2$^{220flox/220flox}$*) were generated by the Animal Model Core at the University of North Carolina at Chapel Hill using CRISPR/

Cas9-mediated integration of a targeting vector into mouse embryonic stem (ES) cells, followed by ES cell injection into blastocytes and production of chimeric progeny. In brief, the CCTop website (https://crispr.cos.uni-heidelberg.de) was used to identify potential Cas9 guide RNAs targeting *Ank2* intron 35 and the 5' end of exon 37. Selected guide RNAs were cloned into a T7 promoter vector followed by in vitro transcription and spin column purification. Functional testing was performed by transfecting Cas9 protein/guide RNA ribonucleoprotein complexes into a mouse embryonic fibroblast cell line. The guide RNA target regions were amplified from transfected cells and analyzed by T7endo1 assay (NEB) to detect genome editing activity at the target site. Guide RNAs selected for genome editing in mouse embryonic stem cells were *Ank2*-i35-sg73T (protospacer sequence 5'- GGTTCTAGTCTTCCCG A –3') and *Ank2*-E37-sg79B (protospacer sequence 5'- GTCCGGACTTGCTAAGAC –3'). A donor vector was constructed for homologous recombination that included a 1002 bp 5' homology arm corresponding to the sequence immediately 5' of the cut site of Cas9/*Ank2*-i35-sg73T; a LoxP site; 368 bp 3' end of *Ank2* intron 35 including splice acceptor sequence; 3707 bp cDNA encompassing exons 36 and 38–46 (isoform lacking exon 37); a 814 bp stop cassette comprised of 3 tandem copies of SV40 polyadenylation sequence; a FRT-flanked selection cassette with PGK mammalian promoter, a EM7 bacterial promoter, a neomycin resistance gene and PGK polyadenylation cassette, all in reverse orientation relative to the *Ank2* elements; a LoxP site; a second 368 bp 3' end of *Ank2* intron 35 including splice acceptor sequence; a 86 bp segment including 22 bp exon 36 fused to 64 bp 5' end of exon 37; silent point mutations designed to disrupt the *Ank2*-E37-sg79B target site. The sequence GTCTTA was mutated to GTGCTC, corresponding to mutation of a valine codon from GTC to GTG and a leucine codon from TTA to CTC; and a 996 bp 3' homology arm corresponding to sequences immediately 3' of the silent point mutations.

The donor vector was incorporated into C57BL/6 N ES cells by nucleofection with 3 μM Cas9 protein (Thermo Scientific), 1.6 25 μM each Ank2-i35-sg73T and Ank2-E37-sg79B guide RNAs and 200 ng/μl (20 μg total) circular donor vector DNA. Cells were selected on G418 and resistant clones were screened for homologous integration of the donor vector at the *Ank2* locus. PCR-positive clones were analyzed by Southern blot with 5' and 3' external probes and neomycin cassette internal probe. Two clones, F6 and H10, were identified with homologous integration of the donor. Targeted ES cell clones F6 and H10 were injected in Albino C57BL/6 N blastocysts for chimera production. Chimeras were mated to transgenic animals expressing Flp recombinase on Albino C57BL/6 N genetic background. Germline transmission of the targeted allele was obtained from both clones, although clone H10 gave more germline transmission pups. Clone H10 had an apparent random integration event in addition to the homologous event. Therefore, pups were screened to identify clones with the homologous integration event in absence of the random integration event. Selected founders were bred for five generations to C57BL/6 J mice after which heterozygous carriers of the *Ank2* targeting allele (*Ank2*<sup>220flox/+</sup>) were bred to each other to generate homozygous carriers (*Ank2*<sup>220flox/220flox</sup>). The WT *Ank2* (*Ank2*<sup>+</sup>) allele was identified by PCR using primers ABCS-WT-F (5'-GCTTTGTTGTATGTATGAAT GTGCTAC-3') and ABCS-WT-R (5'-TTCCTCATCGCTGACAATAACC-3'), which produce a 328 bp DNA fragment. The *Ank2*<sup>220flox</sup> allele was detected by PCR using primers ABCS-RE-F2 (5'-GCTTGGCTGTGT TCACAAACA-3') and ABCS-RE-R2 (5'-GACTTGCGAGCACAGGAACTT-3'), which produce a 639 bp DNA fragment.

## Plasmids

Plasmids used for transfections include pmCherry-C1 (Clontech), pLAMP1-mGFP (Addgene #34831, gift from Dr. Esteban Dell'Angelica), pmCherry-Lifeact-7 (Addgene #54491, gift from Dr. Michael Davidson), pcDNA3.1-L1CAM (Addgene # 12307, gift from Dr. Vance Lemmon), and pCAG-GFP (Addgene # 16664, gift from Dr. Fred Gage). Plasmids pBa-AnkB440-Halo, pBa-AnkB440-PSK-Halo, pBa-AnkB440<sup>P1843S</sup>-Halo, pBa-AnkB440<sup>E3429V</sup> (gifts from Dr. Vann Bennett) have been previously described (*Yang et al., 2019*; *Chen et al., 2020*). The pBa-AnkB440<sup>R1145Q</sup>-Halo plasmid was generated via GeneArt site-directed mutagenesis (Life Technologies) using primers R1145Q_F (5'-CGCATCAT CACCCAAGACTTCCCACAG-3') and R1145Q_R (5'- CTGTGGGAAGTCTTGGGTGAT GAT GCG-3'). To generate pCAG-AnkB220-GFP, an XhoI and NotI fragment containing the AnkB220 cDNA was cloned into the same sites of pCAG-eGFP-N1. The GFP-tagged AnkB440 vector was generated by digesting pBa-AnkB440-Halo with NheI and MfeI to excise the C terminal Halo tag. GFP was amplified with complementary ends from pCAG-AnkB220-GFP using high fidelity Phusion polymerase

(Takara) and primers F-NheI: 5'-CAACAATGAGGCTAGCCGGGATCCACCGGTCGCC-3' and R-MfeI: 5'- TTAACAACAACAATTGATCTAGAGTCGCGGCCGC-3'. Linearized fragments were assembled using In-Fusion Cloning reagents (Takara). All plasmids were verified by full-length sequencing prior to transfection.

## Antibodies and Fluorescent Dyes

Affinity-purified rabbit pan anti-AnkB and anti-AnK440 antibodies used at a 1:500 dilution for immunohistochemistry and 1:5,000 for western blot, were generated by Dr. Vann Bennett laboratory and have been previously described (*Lorenzo et al., 2014*; *Yang et al., 2019*; *Chen et al., 2020*). Other antibodies used for western blot analysis included rabbit anti-Sema 3 A (1:500, #ab23393), rabbit anti-Nrp1 (1:1,000, # ab25998), rat anti-MBP (1:500, #ab7349), mouse anti-L1CAM (1:200, clone 2C2, # ab24345) all from Abcam. We also used rabbit anti-Plexin A1 (1:200, # APR-081) from Alomone; mouse anti-GluR1 (1:500, clone N355/1, #75–327) from Antibodies Incorporated; rabbit anti-phospho-Cofilin (Ser3) (1:1,000, #3311), rabbit anti-LIMK1 (1:1,000, #3842), rabbit anti-MAG (1:1,000, clone D4G3, #9043), rabbit anti-MOG (1:1,000, clone E5K6T, #96457), and mouse anti-GFAP (1:1,000, clone GA5, #3670) from Cell Signaling; mouse anti-cofilin (1:1,000, clone 1G6A2, #66057–1-Ig) from Proteintech; mouse anti-actin (1:2,000, clone C4, #MAB1501), and mouse anti-CNPase (1:500, #C5922) from Millipore-Sigma and rabbit anti-myelin PLP (1:1,000, #NBP1-87781) from Novus Biologicals. Commercial antibodies used for immunofluorescence included mouse anti-neurofilament (1:200, clone SMI-312, #837904) from Biolegend; mouse anti-βIII-tubulin (1:100, clone TU-20, #MAB1637), rat anti-L1CAM (1:200, clone 324, #MAB5272), guinea pig anti-NeuN (1:1,000, clone A60, #ABN90P) from Millipore-Sigma; sheep anti-alpha/beta tubulin (1:200, #ATN02) from Cytoskeleton; chicken anti-GFP (1:1000, #GFP-1020) from Aves; rabbit anti-Olig2 (1:200, # 13999–1-AP) from Proteintech, and mouse anti-Satb2 (1:200, clone SATBA4B10, # ab51502), rat anti-Ctip2 (1:500, clone 25B6, # ab18465), rat anti-MBP (1:100, #ab7349) and rabbit anti-Tbr1 (1:200, # ab31940), all from Abcam. Detection of surface Nrp1 by immunofluorescence was conducted using rabbit anti-Nrp1 extracellular antibody (1:50, # ANR-063, Alomone) that recognizes amino acid residues 502–514 in the extracellular N-terminus of Nrp1. The presence of the Halo tag was detected by incubation with the JF 549 HaloTag ligand from Janelia Farm. F-actin was labeled with phalloidin conjugated to Alexa Fluo-647, 568, and 488 dyes (1:Life Technologies). Secondary antibodies purchased from Life Technologies were used at 1:400 dilution for fluorescence-based detection by confocal microscopy. Secondary antibodies included donkey anti-rabbit IgG conjugated to Alexa Fluor 568 (#A10042), donkey anti-mouse IgG conjugated to Alexa Fluor 488 (#A21202), goat anti-chicken IgG conjugated to Alexa Fluor 488 (#A11039), donkey anti-rat IgG conjugated to Alexa Fluor 647 (#A21247), goat anti-rat IgG conjugated to Alexa Fluor 568 (#A11077), donkey anti-mouse IgG conjugated to Alexa Fluor 568 (#A10037), donkey anti-rabbit IgG conjugated to Alexa Fluor 647 (#A31573), goat anti-rabbit IgG conjugated to Alexa Fluor 594 (#R37117), goat anti-guinea pig IgG conjugated to Alexa Fluor 647 (#A-21450) and goat anti-mouse IgG conjugated to Alexa Fluor 488 (#A11001).Fluorescent signals in western blot analysis were detected using goat anti-rabbit 800CW (1:15000, #926–32211), goat anti-mouse 680RD (1:15000, #926–68070) and goat anti-rat 680RD (1:15000, # 926–68076) from LiCOR.

## Cell Lines

Experiments were conducted in HEK 293T/17 cells (ATCC CRL-11268). This cell line was authenticated by ATCC based on its short tandem repeat profile and have recurrently tested negative for mycoplasma contamination.

## Plasmid Transfection for Biochemistry Analysis

Transfection of Halo-tagged AnkB440 plasmids were conducted in HEK 293T/17 cells grown in 10 cm culture plates using the calcium phosphate transfection kit (Takara) and 8 μg of plasmid. Cell pellets were collected 48 hours after transfection.

## Inmunoblots

Protein homogenates from mouse brains or transfected cells were prepared in 1:9 (wt/vol) ratio of homogenization buffer (8 M urea, 5 % SDS (wt/vol), 50 mM Tris pH 7.4, 5 mM EDTA, 5 mM N-ethylmeimide, protease and phosphatase inhibitors coktails) and heated at 65 °C for 15 min to produce a

clear homogenate. Total protein lysates were mixed at a 1:1 ratio with 5 x PAGE buffer (5 % SDS (wt/vol), 25 % sucrose (wt/vol), 50 mM Tris pH 8, 5 mM EDTA, bromophenol blue) and heated for 15 min at 65 °C. Protein lysates from DIV3 OPCs were prepared by washing cultured oligodendrocytes in PBS. Cells were then scraped with a cell lifter and then centrifuged at >15,000 g. The cell pellet was lysed on ice for 20 min in a lysis buffer containing HEM (50 mM HEPES, 1 mM EDTA, 1 mM MgCl2), 25 mM NaCl, 0.5% TX-100, and protease inhibitor cocktail (Sigma P8340). All lysates were resolved by SDS-PAGE on 3.5%–17.5% acrylamide gradient gels in Fairbanks Running Buffer (40 mM Tris pH 7.4, 20 mM NaAc, 2 mM EDTA, 0.2 % SDS (wt/vol)). Proteins were transferred at 29 V overnight onto 0.45 µm nitrocellulose membranes (#1620115, BioRad) at 4 °C in methanol transfer buffer (25 mM Tris, 1.92 M Glycine, 20 % methanol). Transfer efficiency was determined by Ponceau-S stain. Membranes were blocked in TBS containing 5 % non-fat milk for 1 hour at room temperature and incubated overnight with primary antibodies diluted in antibody buffer (TBS, 5 % BSA, 0.1 % Tween-20). After three washes in TBST (TBS, 0.1 % Tween-20), membranes were incubated with secondary antibodies diluted in antibody buffer for two hours at room temperature. Membranes were washed 3 x for 10 minutes with TBST and 2 x for 5 minutes in TBS. Protein-antibody complexes were detected using the Odyssey CLx Imaging system (LI-COR).

### Labeling and Detection of Biotinylated Surface Proteins

Cortical neuronal cultures were washed three times with ice-cold PBSCM (PBS + 1 mM MgCl$_2$ +0.1 mM CaCl$_2$) and incubated with 0.5 mg/ml Sulfo-NHS-SS-biotin (Life Technologies) for 1 hour at 4 °C. Reactive biotin was quenched by two consecutive 7 minute incubations with 20 mM glycine in PBSCM on ice. Cell lysates were prepared in TBS containing 150 mM NaCl, 0.32 M sucrose, 2 mM EDTA, 1 % Triton X-100, 0.5% NP40, 0.1 % SDS, and complete protease inhibitor cocktail (Sigma). Cell lysates were incubated with rotation for 1 hour at 4 °C and centrifuged at 100,000 x g for 30 min. Soluble fractions were collected and incubated with high capacity NeutrAvidin agarose beads (Pierce) overnight at 4 °C to capture biotinylated surface proteins. Beads were washed three times with TBST. Proteins were eluted in 5x-PAGE buffer and resolved by SDS-PAGE and western blot.

### Histology and Immunohistochemistry

Brains were fixed by transcardial perfusion with 4 % PFA in PBS before overnight immersion in the same fixative at 4 °C. After fixation, brains were stored in PBS at 4 °C until use. Brains were then transferred to 70 % ethanol for 24 hours and paraffin embedded. 10 µm coronal brain sections were cut using a microtome (Leica RM2135) and mounted on glass slides. Sections were deparaffinized and rehydrated using a standard protocol of washes: 3 × 3 xylene washes, 3 × 2 min 100 % ethanol washes, and 1 × 2 min 95%, 80%, 70%, ethanol each, followed by ≥5 min in PBS. Sections were processed for antigen retrieval using 10 mM sodium citrate with 0.5 % Tween-20, pH six in a pressure cooker for 3 minutes at maximum pressure. Sections were cooled, washed in PBS and blocked in antibody buffer (4 % BSA, 0.1 % Tween-20 in PBS) for 90 minutes at room temperature. Tissue sections were then incubated with primary antibody in antibody buffer overnight at 4 °C and with secondary antibodies for 2 hours at room temperature, washed with PBS, incubated with DAPI where applicable, and mounted with Prolong Gold Antifade Reagent (Life Technologies). For frozen sections, brains were fixed as above, sequentially incubated in 5%, 15% and 30% sucrose solutions in PBS at 4 °C. Cryoprotected brains were embedded in Epredia Cryomatrix embedding resin (Fisher). 10 µm frozen coronal brain sections, obtained using a CryoStar NX50 Cryostat (Fisher Scientific), were permeabilized in 0.2 % Triton X-100 in PBS for 30 minutes at room temperature and stained as above. 100 µm coronal brain sections, obtained from 4 % PFA-fixed brains using a VT100S vibratome (Leica), were permeabilized in 0.3 % Triton X-100 in PBS for two hours at room temperature and processed for immunofluorescence staining as above, with the exception of addition of 0.3 % Triton to blocking and antibody buffers.

### Transmission Electron Microscopy

Control (*AnkB440$^{flox/flox}$*) and AnkB440 KO (*AnkB440$^{flox/flox}$::Nestin-Cre*) mice were perfused at PND25 with 2 % PFA and 2.5 % glutaraldehyde in 0.1 M sodium cacodylate buffer (pH 7.4) at room temperature. Brain were post-fixed in the same buffer at 4 °C for three additional days. Corpus callosum sections were micro-dissected, then treated with 1 % OsO4 in cacodylate buffer for 1 h on ice and 0.25 % uranyl acetate in acetate buffer at pH5 overnight at 4 °C. Sections were then dehydrated with

ethanol and embedded in epoxy. TEM images were acquired with a JEOL 1230 transmission electron microscope at the Microscopy Service Laboratory at UNC-Chapel Hill and with a JEOL 1400 at the NIH-NINDS electron microscopy core. The g-ratio of myelinated corpus callosum axons and the percent of total and myelinated corpus callosum axons was evaluated using ImageJ.

## Golgi Stain

Golgi staining of PND25 brains was conducted using the FD Rapid GolgiStain Kit (FD Neurotechnologies Inc) In brief, brains were immersed in Solutions A + B for 2–3 weeks, before being transferred to solution C for 3–6 days. 100 µm coronal cryosections from areas of the somatosensory cortex were collected on gelatin-coated microscope slides, counter-stained following manufacturer recommendations, and mounted in Permount for imaging.

## In Utero Electroporation

Lateral ventricles of E15.5 *AnkB440*$^{flox/flox}$ and *AnkB440*$^{flox/flox}$::Nestin-Cre embryos were electroporated as previously reported (*Guo et al., 2019*). 1 µl of pCAG-GFP DNA (2 µg/µl) prepared using the EndoFree Plasmid kit (Qiagen) was mixed with Fast Green and injected into the lateral ventricles. DNA was electroporated by delivering 5 pulses at 30 V for 50 ms at 950 ms intervals through the uterine wall using a BTX ElectroSquarePorator (ECM 830). Embryos were allowed to develop and brains were collected at PND25 for histological evaluations.

## Oligodendrocyte Primary Cell Culture

Oligodendrocyte precursor cells (OPCs) were purified from Sprague-Dawley rat pups (P6–P8) by immunopanning as previously described (*Emery and Dugas, 2013*). Briefly, cortical tissue was dissociated by papain digestion and filtered through a Nitex mesh to obtain a mixed single-cell suspension. This suspension was incubated in two negative-selection plates coated with anti-Ran-2 and anti-GalC antibodies, then in a positive-selection plate coated with anti-O4 antibody. Adherent cells were trypsinized, then differentiated in media containing T3 (thyroid hormone).

## RNA Isolation and qPCR Analysis

RNA was isolated from PND25 control mouse brains, DIV17 mouse astrocyte cultures, DIV3 rat OPC cultures, and from 4 month old control mouse skeletal muscle and brown adipose tissue using the RNAeasy extraction kit (Qiagen) following manufacturer's instructions. cDNA libraries prepared using the SuperScript IV VILO cDNA synthesis kit (ThermoFisher Scientific). qPCR analysis were performed using 100 ng of each cDNA and PowerUp SYBR Green Master Mix (ThermoFisher Scientific) in a QuantStudio 7 Flex Real-Time PCR System. MBP transcript levels were detected with primers MBP-3'UTR-F (CGACACCTCCTTATCCCTCTAA) and MBP-3'UTR-R (TTGGGCTACATCAACCATCAC). AnkB440 transcripts were detected using primers AnkB440-F (GCCCTGGAAGTGTCAGTCAT) and AnkB440-R (AAGCCAGCCTCTCTTCCATC). Total AnkB transcripts were detected using primers panAnkB-F (CGGAGTCCGATCAAGAGCAG) and AnkB440-R (AAGCCAGCCTCTCTTCCATC). Transcript levels were normalized to internal GAPDH mRNA expression detected using GAPDH-F (TCAACGGGAAGCCCATCA) and GAPDH-R (CTCGTGGTTCACACCCATCA) primers. RQ (relative quantification) for each transcript in each tissue was computed using the $\Delta\Delta$Ct method and expressed relative to the transcript expression in brain.

## Primary Neuron Culture

Primary cortical neuronal cultures were established from E16-PND0 mice. Cortices were dissected in Hibernate E (Life Technologies) and digested with 0.25 % trypsin in HBSS (Life Technologies) for 20 min at 37 °C. Tissue was washed three times with HBSS, dissociated in DMEM (Life Technologies) supplemented with 5 % fetal bovine serum (FBS, Genesee), and gently triturated through a glass pipette with a fire-polished tip. Dissociated cells were filtered through a 70 µm cell strainer to remove any residual non-dissociated tissue and plated onto poly-D-lysine-coated 1.5 mm coverglasses or dishes (MatTek) at a density of $4 \times 10^4$ cells/cm² for transfection and imaging experiments. For all cultures, media was replaced 3 hours after plating with serum-free Neurobasal-A medium containing B27 supplement (Life Technologies), 2 mM Glutamax (Life Technologies), and 1 % penicillin/streptomycin (Life Technologies) (neuronal growth media). 5 µM cytosine-D-arabinofuranoside (Sigma) was

added to the culture medium to inhibit the growth of glial cells three days after plating. Neurons were maintained at 37 °C with 5 % $CO_2$ and fed twice a week with freshly made culture medium until use.

## Plasmid Transfection for Time-Lapse Live Imaging and Immunofluorescence Analysis

For neuronal rescue experiments, AnkB440 KO primary cortical neurons were transfected with Halo-tagged AnkB440 or GFP-tagged AnkB220 plasmids at DIV0 by lipofection (Lipofectamine 2000, Thermo Fisher Scientific). After brain dissociation, $2.5 \times 10^5$ cells/ml neurons were incubated with Lipofectamine 2000 (12 µl/ml) and plasmid DNA (2.5 µg/ml) in suspension at 37 °C for 45 min. Cells were pelleted at 200 g for 4 minutes and plated in poly-D-lysine-coated coverglasses or dishes and used for studies at 16 hours after transfection and plating (DIV1), 28–36 hours post plating (DIV1-2), 72 hours post plating (DIV3), or five days post plating (DIV5). For time-lapse imaging of organelle dynamics, DIV5 cortical neurons were co-transfected with 1 µg of pLAMP1-GFP using lipofectamine 2000 and imaged 48–96 hours after transfection. For experiments that evaluate axonal length and branching, DIV0 neurons were transfected with 500 ng of pmCherry-C1, either alone or in combination with Halo-tagged AnkB440 or GFP-tagged AnkB220 plasmids. Neurons were processed for immunofluorescence one or five days after transfection. For assessment of actin patch dynamics, neurons were transfected with 500 ng of mCherry-Lifeact plasmid at DIV0 by lipofection and evaluated at DIV1 through time-lapse microscopy. For assessment of actin dynamics during Sema 3A-induced GC collapse, neurons were transfected with 500 ng of mCherry-Lifeact plasmid at DIV0 by lipofection and evaluated at DIV3 through time-lapse microscopy upon treatment with Sema 3 A. Evaluation of L1CAM binding to GFP-tagged AnkB220 or AnkB440 through a membrane recruitment assay was conducted HEK293T in cells independently transfected with 100 ng of pcDNA3.1-L1CAM, GFP-AnkB220, or GFP-AnkB440 plasmids, or co-transfected with 100 ng of pcDNA3.1-L1CAM in combination with either GFP-AnkB220 or GFP-AnkB440 plasmids 48 hours post-transfection.

## Growth Cone Collapse Experiments

DIV3 neurons were treated with Sema 3 A (250 ng/ml, Peprotech, # 150–17 H) or Ephrin A5 (1 µg/ml, R&D Systems, # 374-EA) for 1, 5, 15, or 30 min in neuronal growth media, fixed for 15 min with 4 % PFA/4 % sucrose, and washed in PBS. Neurons were permeabilized with 0.2 % Triton X-100 in PBS for 10 min and blocked in antibody buffer (4 % BSA, 0.1 % Tween-20 in PBS) for 60 min at room temperature. Neurons were stained with primary antibodies overnight at 4 °C and with a mix of secondary antibodies and phalloidin for two hours at room temperature. Neurons transfected with plasmids expressing Halo-tagged AnkB440 proteins were treated with fresh media containing Sema 3 A and with the JF 549 HaloTag ligand (1:200) for 30 min. The percent of collapsed growth cones was recorded for each experimental condition, selecting for transfection when applicable. Axon growth cones were recorded as collapsed based on the presence of a pencil-like shape devoid of lamellae and with a maximum of one filopodia, or intact based on the fan-shaped morphology enriched in filopodia and/or lamellipodia.

## Immunocytochemistry

Neuron and OPC cultures cells were grown in PDL hydrobromide-coated 1.5 glass coverslips. HEK293T cells were grown in uncoated MatTek dishes. All cells were washed with cold PBS, fixed with 4 % PFA for 10 min, and permeabilized with either 0.2 % Triton-X100 in PBS for 10 min (neurons and HEK293T cells) or with 0.1 % Triton X-100 in PBS for 3 min at room temperature. Cells were blocked with 5 % donkey serum with 1 % BSA in PBS for 30 min at room temperature and processed for staining as tissue sections. For F-actin labeling, Alexa Fluor 488-, Alexa Fluor 568-, or Alexa Fluor 633-conjugated phalloidin (1:100) was added to the secondary antibody mix. To label surface Nrp1, fixed but non-permeabilized DIV3 cortical neurons, untreated and treated with Sema 3 A, were blocked as described above and incubated overnight at 44 °C with a rabbit anti-Nrp1 antibody that recognizes an extracellular epitope, followed by overnight incubation with donkey anti-rabbit IgG conjugated to Alexa Fluor 568. Neurons were fixed again for 10 min with 4 % PFA in PBS, permeabilized for 8 minutes with 0.2 % Triton X-100 in PBS, reblocked with antibody buffer for 30 minutes at room temperature and incubated with primary antibodies overnight at 4 °C. Secondary antibodies and phalloidin were

applied in antibody buffer overnight at °4 C. Cells were mounted for imaging with Prolong Gold Antifade Reagent (Life technologies).

## Proximity Ligation Assay (PLA)

PLA was performed using the commercial Duolink kit (Sigma-Aldrich) following the manufacturer's recommendations. Fixed and permeabilized neurons were incubated overnight with a pair of primary antibodies specific for the putative interacting partners, each produced in different species. Duolink minus- and plus-probes were used to detect antibody-labeled proteins. In the case of the detection of PLA signal between surface Nrp1 and L1CAM, fixed, but non-permeabilized neurons were first incubated overnight at 4 °C with a rabbit anti-Nrp1 antibody that recognizes an extracellular epitope, followed by cell permeabilization and overnight incubation with mouse anti-L1CAM.

## Image Acquisition and Image Analysis

Brain sections and cultures cells stained with antibodies were imaged using a Zeiss 780 laser scanning confocal microscope (Zeiss) and 405-, 488-, 561-, and 633 nm lasers. Single images and Z-stacks and tile scans were collected using the 10 x (0.45 NA), 20 x (0.8 NA) and Plan Apochromat 40 x oil (1.4 NA) and 63 x oil (1.4 NA) objective lenses. Images of OPCs were acquired on a spinning-disk confocal microscope (Nikon Ti2 inverted body with Yokogawa X1 spinning disk) using a scMOS camera (Hamamatsu FusionBT) run by Nikon Elements software. Images were processed, and measurements taken and analyzed using NIH ImageJ software. Three-dimensional rendering of confocal Z-stacks was performed using Imaris (Bitplane). Golgi-stained brains were imaged using a Nikon Ti2 Eclipse scope running NIS-Elements. Widefield Z-stacks were taken with 2 μm optical section in the primary somatosensory cortex with a 40 x/NA objective. To quantify CC fasciculation and targeting of contralateral projections, lines were traced through indicated regions of interest and measurements of fluorescence intensity obtained using the "Plot profile" function of ImageJ. AnkB440 staining intensity in OPCs was analyzed by drawing a circle region of interest (ROI) around the cell and quantifying the intensity within this ROI using Nikon Elements software.

## Time-Lapse Video Microscopy and Movie Analyses

Live microscopy of neuronal cultures was carried out using a Zeiss 780 laser scanning confocal microscope (Zeiss) equipped with a GaAsP detector and a temperature- and $CO_2$-controlled incubation chamber as previously reported (*Snouwaert et al., 2018*). Movies were taken in the distal axon and captured at a rate of 1 frame/second for time intervals ranging from 60 to 300 seconds with a 40 x oil objective (1.4NA) using the zoom and definite focus functions. Movies were processed and analyzed using ImageJ (http://rsb.info.nih.gov/ij). Kymographs were obtained using the KymoToolBox plugin for ImageJ (https://github.com/fabricecordelieres/IJ_KymoToolBox; *Zala et al., 2013*). In details, space (x axis in μm) and time (y axis in sec) calibrated kymographs were generated from video files. In addition, the KymoToolBox plugin was used to manually follow a subset of particles from each kymograph and report the tracked particles on the original kymograph and video files using a color code for movement directionality (red for anterograde, green for retrograde and blue for stationary particles). Quantitative analyses were performed manually by following the trajectories of individual particles to calculate dynamic parameters including, net and directional velocities and net and directional run length, as well as time of pause or movement in a direction of transport. Anterograde and retrograde motile vesicles were defined as particles showing a net displacement >3 μm in one direction. Stationary vesicles were defined as particles with a net displacement <2 μm.

## Statistical analysis

Sample size (n) for evaluations of growth cone collapse was estimated using power analyses and expected effect sizes based on preliminary data in which we used similar methodologies, specimens, and reagents. We assumed a moderate effect size ($f$ = 0.25–0.4), an error probability of 0.05, and sufficient power (1-β = 0.8). GraphPad Prism (GraphPad Software) was used for statistical analysis. Two groups of measurements were compared by unpaired, two tailed students $t$-test. Multiple groups were compared by one-way ANOVA followed by Tukey or Dunnett's multiple comparisons tests.

## Acknowledgements

We thank Dr. Vann Bennett for the gift of the total AnkB KO, AnkB440 KO and L1CAM[Y1229H] mice and Dr. Matthew Rasband for the gift of the βII-spectrin KO mice. We also that Dr. Dale Cowley and the Animal Model Core at the University of North Carolina at Chapel Hill for the generation of *Ank2*[220flox] mice. TEM sample preparation was performed with assistance from Susan Cheng and Sandra Lara (NINDS electron microscopy core) and Kristen White (University of North Carolina at Chapel Hill Microscopy Service Laboratory). DNL was supported by the University of North Carolina at Chapel Hill School of Medicine as a Simmons Scholar and by the US National Institutes of Health (NIH) grant R01NS110810. ESA and CRC were supported by NIH grants NS116859 and HD098657. MMF, ED, JR were supported by the Intramural Research Program of NINDS (NIH). Microscopy performed at the Neuroscience Microscopy Core Facility was supported, in part, by funding from the NIH-NINDS Neuroscience Center Grant P30 NS045892 and the NIH-NICHD Intellectual and Developmental Disabilities Research Center Support Grant U54 HD079124.

## Additional information

### Funding

| Funder | Grant reference number | Author |
| --- | --- | --- |
| National Institute of Neurological Disorders and Stroke | NS110810 | Damaris N Lorenzo<br>Blake A Creighton<br>Simone Afriyie<br>Deepa Ajit |
| National Institute of Neurological Disorders and Stroke | P30 NS045892 | Damaris N Lorenzo |
| Eunice Kennedy Shriver National Institute of Child Health and Human Development | U54 HD079124 | Damaris N Lorenzo |
| National Institute of Neurological Disorders and Stroke | NS116859 | Cristine R Casingal<br>ES Anton |
| Eunice Kennedy Shriver National Institute of Child Health and Human Development | HD098657 | Cristine R Casingal<br>ES Anton |

The funders had no role in study design, data collection and interpretation, or the decision to submit the work for publication.

### Author contributions

Blake A Creighton, Formal analysis, Funding acquisition, Supervision, Investigation; Simone Afriyie, Cristine R Casingal, Kayleigh M Voos, Joan Reger, Formal analysis, Funding acquisition; Deepa Ajit, Formal analysis, Funding acquisition, Investigation; April M Burch, Eric Dyne, Julia Bay, Funding acquisition; Jeffrey K Huang, Funding acquisition, Supervision; ES Anton, Formal analysis, Funding acquisition, Supervision; Meng-Meng Fu, Formal analysis, Funding acquisition, Funding acquisition, Supervision, Supervision, Investigation; Damaris N Lorenzo, Conceptualization, Data curation, Formal analysis, Funding acquisition, Funding acquisition, Supervision, Project administration, Supervision, Writing - original draft, Investigation

### Author ORCIDs

Deepa Ajit http://orcid.org/0000-0002-6062-7070
Damaris N Lorenzo http://orcid.org/0000-0002-6856-2988

### Ethics

This study was performed in strict accordance with the recommendations in the Guide for the Care and Use of Laboratory Animals of the National Institutes of Health. All of the animals were handled

according to approved institutional animal care and use committee (IACUC) protocol (#19-209) of the University of North Carolina at Chapel Hill.

## Decision letter and Author response

Author response https://doi.org/10.7554/eLife.69815.sa2

---

## Additional files

### Supplementary files

• Transparent reporting form

### Data availability

All data generated or analyzed during this study are included in the manuscript and supporting files. Source data files have been provided for Figures 1–9 and Figure 1—figure supplements 1–4, Figure 2—figure supplement 1, Figure 3—figure supplements 1 and 2, Figure 5—figure supplements 1 and 2, Figure 6—figure supplement 1, and Figure 7—figure supplement 1. Additional microscopy datasets have been deposited in Dryad.

The following dataset was generated:

| Author(s) | Year | Dataset title | Dataset URL | Database and Identifier |
|---|---|---|---|---|
| Lorenzo D | 2021 | AnkB440 GC collapse | https://doi.org/10.5061/dryad.np5hqbzsz | Dryad Digital Repository, 10.5061/dryad.np5hqbzsz |

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
