## [Editor Report]

In their paper, Creighton et al. investigate the mechanisms regulating cytoskeletal changes mediating neuronal branching and axon growth. They assess the role of the neuronal specific form of the scaffolding protein ankyrin-B, and how its depletion or autism-spectrum disorder mutations affect cortical axon branching, targeting, and developmental refinement. Overall, their work develops a new model for how ankyrin-B functions with the Sema 3A receptor complex formed by cell adhesion molecule L1CAM and neuropilin-1, and an actin severing protein (cofilin) during cortical axon development and provides insight into how this process is altered in autism spectrum disorders. This manuscript will be of interest to cell, developmental, and neuronal biologists interested in the biological mechanisms of early circuit development, and how these mechanisms relate to autism spectrum disorders.

---

## [Decision Letter]

**Decision letter after peer review:**

Thank you for submitting your article "Giant ankyrin-B mediates transduction of axon guidance and collateral branch pruning factor Sema 3A" for consideration by *eLife*. Your article has been reviewed by 3 peer reviewers, and the evaluation has been overseen by a Reviewing Editor and Anna Akhmanova as the Senior Editor. The following individual involved in review of your submission has agreed to reveal their identity: Linda J Richards (Reviewer #1).

Essential revisions:

1) All of the reviewers agreed that the authors should provide more data and analyses to explain the increase in corpus callosum thickness in the AnkB440 mutant mice. The reasoning behind the corpus callosum enlargement is unclear and the reviewers suggest a number of experiments that can be performed to address this interesting observation.

2) All of the reviewers agreed that additional in vivo experiments should be performed to analyze other phenotypes of the AnkB440 mice. Are other axonal tracts increased in size or only the corpus callosum? Is myelination normal? Are the general trajectories of the cortical axons impaired? Do cultured AnkB440 mutant neurons extend a similar number of processes as wild type neurons before axon specification (presumably the authors already have this data that they can perform additional analyses on)? The results of these experiments would be very informative for this study.

3) The reviewers suggested that the authors further examine the effects of mutant AnkB440 on certain functions of the Sema3 signaling pathway. For example, is the contralateral targeting of callosal axons perturbed in the AnkB440 mutant mice (this links back to #1 as well)?

4) The reviewers suggested that the authors flesh out the role of AnkB a bit more with regards to its effects on F-actin. For example, in cultured AnkB440 mutant neurons, does the difference in F-actin patches impact process initiation or stability or both?

5) One reviewer suggested adding a working model at the end, which would nicely tie together the data and overall conclusions of the paper. Due to the involvement of AnkB in Sema3A signaling and subsequent actin remodeling, we highly recommend a cartoon to help guide the reader.

6) Please refer to the specific comments that the reviewers make about image quality, inclusion of replicate numbers and relevant statistical tests, figure labels and keys, and general text editing to improve the overall accessibility of the manuscript.

*Reviewer #1 (Recommendations for the authors):*

It would be helpful to provide more information about the in vivo phenotype. Are other axon tracts increased in size? Yang et al., 2019 identified other axonal tracts were also enlarged in AnkB440 KO mice. Corpus callosum hyperplasia is seen in patients with neurofibromatosis 1. In this condition, it appears to be due to alterations in oligodendrocyte development and myelination (Wang et al., 2012 Cell doi: 10.1016/j.cell.2012.06.034; Asleh et al. 2020; DOI: 10.1073/pnas.2008391117). Are the authors certain that AnkB440 is not expressed in oligodendrocytes? Is myelination normal in the mutants or is there increased wrapping of the axons? If AnkB440 is only expressed by neurons, it could be that there is an effect on activity of the callosal axons which then affects oligodendrocytes in a non-cell-autonomous manner (see Mitew et al., 2018 Nature Comms doi: 10.1038/s41467-017-02719-2). Nrp1-Sema3A signalling is also required for the targeting of callosal axons in the contralateral hemisphere (Zhou et al., 2013 PNAS, doi: 10.1073/pnas.1310233110). It is possible that contralateral targeting of callosal axons and/or their selection of contralateral targets might be disrupted in AnkB440 mutants. This could be examined using in utero electroporation of a reporter construct to achieve sparse axonal labelling to examine the entire callosal trajectory. This would provide more compelling in vivo evidence for increased axonal branching than is presented with the Golgi staining of cell bodies with increased dendritic branching.

The poor image contrast in Figure 1 makes it difficult to identify the filopodia in the image. The figure legend mentions asterisks for panel D but there are none on the image.

*Reviewer #2 (Recommendations for the authors):*

I have several questions regarding the consequences of ANKB440 deletion on cortical connectivity. It is not clear to me how the authors link the excess branching and the defective response to Sema3A to alterations of cortical connectivity. For example, the authors report an increase of the corpus callosum thickness in ANKB440 mutants. With their analysis of the local branching pattern of cortical neurons with Golgi staining, do the authors want to suggest that CC thickness increase is due to additional collaterals that would have traveled along with the primary axon? Do they rather think that ANKB440 cortical axons elaborate more collaterals from their shaft during their navigation?

Could CC thickness increase be due to axon-axon defasciculation? L1-CAM- Nrp1 labeling could be achieved to look at how fibers are organized in the CC tract.

Is it feasible to distinguish the response of the growth cone of the main axon from that of the growth cones from the collaterals? Do the authors think that the lack of response to Sema3A of ANKB440 cortical neurons will impact in vivo on the pruning or rather on the initial number of formed collaterals?

– Given the defective response to a guidance cue, cortical axons could select wrong trajectories. Do the authors think this is the case? This could be examined at developmental stages with DiI tracing.

– Is there a reason thinking that defects in the mutant are restricted to the corpus callosum?

– Is ANKB440 deficit impacts on the early polarity of cortical neurons. In culture, do the neurons extend similar number of processes before axon specification? Is the observed difference of F-actin patches in context of ANKB440 deletion expected to impact on process initiation?

– How could they explain the maintained sensitivity to EphrinA5, since it was reported that L1-CAM is required for the response to EphrinA5 (Demyanenko et al., 2011)?

– Do the authors think that alteration of active/inactive cofilin ratio could have additional outcome in the mutant?

– The authors should indicate more clearly in the manuscript, or figures the number of replicates, the number of analyzed cases for each experiment. They should also indicate in the figure panels the developmental stage at which the observations were made.

*Reviewer #3 (Recommendations for the authors):*

The interpretation of the results in Figure 2A/B are a bit confusing. Do the authors suggest that the corpus callosum is thicker due to branching before the corpus callosum? In Figure 2E/F they show increased branching near the cell bodies on the ipsilateral side of the cortex, but how would this relate to the callosal defect? Since the difference in callosal thickness is only close to the midline, wouldn't this suggest some defect is specifically occurring there?

Figure 3C – The middle two images (and labels) in the top row of images should be switched to be consistent with the bottom two rows.

The title of Figure 3 is that AnkB440 is enriched at axonal growth cones. This is an overinterpretation. The figure shows that AnkB440 is present in growth cones but it does not appear to be enriched there. To say it is enriched there they would have to show more convincing images and quantify the levels in the growth cone and axon shaft (normalized to cytoplasmic protein) and show enrichment.

Rescue of the AnkB440 KO phenotype is shown in Figure 4E-G with AnkB440-Halo but not AnkB220-GFP, which is a nice experiment. It shows AnkB440 is required for the Sem3A phenotype and it is specific to the neural-specific isoform. Why weren't these experiments carried out for the filopodia and branching experiments in Figure 1D-H and Figure 1 – figure supp. 2?

Figure 4 B,C and D should not be presented as line graphs, as they were not repeated measures but separate experiments for each treatment time.

The experiment in Figure 4 – Figure Supp. 4 with Ephrin-A5 still collapsing AnkB440 KO growth cones shows nice specificity to the lack of response to Sema3A and convincingly indicates the growth cones can collapse, indicating a specific, rather than general defect in growth cone response.

Figure 5B – The horizontal and vertical labels appear to be reversed based on the images of the growth cones.

It is unclear why an unpaired t-test was used in 4G but ANOVA was used in Figure 5C. ANOVA is appropriate for multiple conditions, t-tests are not. Unpaired t-test shouldn't be used in Figure 6B, D, Figure 6-Figure supp. 6, Figure 8F – for multiple conditions.

The keys are confusing in several of the figures. In Figure 2D the dots for the AnkB440 KO are one color in the key but different in the graph. In Figure 4G the bars are in the key, not the dots. In Figures 5C/E, 6H and 7E the key is again the color of the dots, not the bars. Also, the dots for AnkB440 KO and L1CAM-Y1229H are very similar color, making it confusing. To make it easier to interpret, the bar color, not the dot color, should be used in the keys.

Could the authors include a working model at the end? How is AnkB440 functioning with L1CAM, Nrp1 and cofilin to influence the actin cytoskeleton in response to Sema3A? The authors have put together many very nice experiments but in the discussion it was hard to put together a mechanistic model of how AnkB440 is functioning in growth cone collapse/branch inhibition.

---

## [Author Response]

Reviewer #1 (Recommendations for the authors):It would be helpful to provide more information about the in vivo phenotype. Are other axon tracts increased in size? Yang et al., 2019 identified other axonal tracts were also enlarged in AnkB440 KO mice. Corpus callosum hyperplasia is seen in patients with neurofibromatosis 1. In this condition, it appears to be due to alterations in oligodendrocyte development and myelination (Wang et al., 2012 Cell doi: 10.1016/j.cell.2012.06.034; Asleh et al. 2020; DOI: 10.1073/pnas.2008391117).

We thank the reviewer for raising these questions. As a point of clarification, we would like to underscore that in the Yang et al., 2019 manuscript, the reported Diffusion Tensor Imaging data belongs to knock-in AnkB440^R2589fs^ mice (referred to in that publication as ABfs/fs), which models a human ASD-linked *ANK2* variant within the neuronal specific region of AnkB440. As shown in Yang et al., 2019 and in Figure 9A,B of our revised manuscript, the R2589fs AnkB440 variant does not eliminate expression of AnkB440, but rather results in a truncated 290kDa fragment that retains some of the functional domains of the protein. Both AnkB440 KO and AnkB440^R2589fs^ cortical neurons show increased axon branching in vitro (Yang et al., 2019), suggesting a common mechanism. However, the 290kDa R2589fs AnkB440 fragment alters AnkB440 distribution and is additionally likely to result in gain-of function or dominant-negative effects that might impact axon development and cortical connectivity in ways different from total loss of AnkB440. This is partly one of the motivations for assessing for the first time axon tract morphology in AnkB440 KO brains.

Are the authors certain that AnkB440 is not expressed in oligodendrocytes?

We evaluated AnkB440 expression in differentiated oligodendrocyte precursor cells (OPCs) by immunofluorescence staining, western blot, and qPCR. The results of these assays (Figure 3—figure supplement 1) confirm that AnkB440 is not expressed at detectable levels in OPCs. Thus, we ruled out a cell-autonomous effect of AnkB440 loss in oligodendrocytes.

Is myelination normal in the mutants or is there increased wrapping of the axons?

We thank the reviewer for raising this question. We evaluated myelination in the corpus callosum of PND25 control and AnkB440 KO mice by TEM and found no differences in the g-ratio of myelinated axons or in the number of myelinated axons per area (Figure 3A-D). We also did not detect differences in the levels of myelin basic protein (MBP) in the CC, assessed by immunofluorescence staining in AnkB440 KO mice (Figure 3F,G). On the other hand, AnkB440 KO CC axons showed loose myelin wraps (*Figure 3A, open red arrowhead*) and enlarged inner tongues near the axonal membrane (*Figure 3A, open blue arrowheads*). Thus, AnkB440 is likely involved in contact-mediated signals between the axon and myelin wrap that promote myelin ultrastructural organization and health, which when deficient could alter CC axon organization and function.

If AnkB440 is only expressed by neurons, it could be that there is an effect on activity of the callosal axons which then affects oligodendrocytes in a non-cell-autonomous manner (see Mitew et al., 2018 Nature Comms doi: 10.1038/s41467-017-02719-2).

While our studies did not directly evaluate the activity of callosal axons or the density and proliferation of immature OPCs, we found that the total count of CC cells expressing the oligodendrocyte lineage-specific marker Olig2 was preserved and that the density of Olig2+ cells/CC area was reduced in AnkB440 KO brains (Figure 3—figure supplement 2). In addition, loss of AnkB440 did not alter protein levels of MBP and of other proteins associated with oligodendrocyte maturation and myelination (Figure 3H,I).

Nrp1-Sema3A signalling is also required for the targeting of callosal axons in the contralateral hemisphere (Zhou et al., 2013 PNAS, doi: 10.1073/pnas.1310233110). It is possible that contralateral targeting of callosal axons and/or their selection of contralateral targets might be disrupted in AnkB440 mutants. This could be examined using in utero electroporation of a reporter construct to achieve sparse axonal labelling to examine the entire callosal trajectory. This would provide more compelling in vivo evidence for increased axonal branching than is presented with the Golgi staining of cell bodies with increased dendritic branching.

Following the suggestion of the reviewer, we labeled S1 callosal projecting axons with GFP via in utero electroporation at E15.5 and evaluated their topographic order within the CC and contralateral targeting at PND25 (Figure 4). We found that in PND25 AnkB440 KO brains GFP-labeled axons originating from S1 show a more widespread distribution at the CC midline, with a significant number of axons navigating through more dorsal CC paths. Contralateral GFP-labeled projections in AnkB440 KO brains show lack of selective targeting to the S1/S2 border, and instead were widely distributed throughout the contralateral somatosensory cortex, with a notable fraction of axons ectopically projecting to more medial S1 regions and to regions below the S1/S2 border. We also detected lack of specific targeting of contralateral LII/III and LV neurons, with increased branching and innervation of GFP axons through all contralateral cortical layers.

The poor image contrast in Figure 1 makes it difficult to identify the filopodia in the image. The figure legend mentions asterisks for panel D but there are none on the image.

We updated Figure 1 with images derived from time-lapse images with better contrast and revised the corresponding annotations and figure legend. *Reviewer #2 (Recommendations for the authors):*

I have several questions regarding the consequences of ANKB440 deletion on cortical connectivity. It is not clear to me how the authors link the excess branching and the defective response to Sema3A to alterations of cortical connectivity. For example, the authors report an increase of the corpus callosum thickness in ANKB440 mutants. With their analysis of the local branching pattern of cortical neurons with Golgi staining, do the authors want to suggest that CC thickness increase is due to additional collaterals that would have traveled along with the primary axon? Do they rather think that ANKB440 cortical axons elaborate more collaterals from their shaft during their navigation?

As we show in Figure 6, the response to Sema 3A is affected in growth cones of both the primary axon and collateral branches in AnkB440 KO neurons, which likely affects targeting and refinement of primary axons and collaterals in vivo. Our expanded findings also show increases in the thickness and disorganization of multiple axonal tracts in AnkB440 KO brains, not only in the CC. In addition, we observe an increase in the number of axonal processes in the CC of PND25 AnkB440 KO brains (Figure 4), which could result from a higher density of axons entering and traveling through the CC and/or from deficits in the developmental pruning of CC axons.

We observe a more widespread dorsoventral distribution of S1 callosal axons at the midline CC, with a significant number of axons navigating through more dorsal CC paths. While developmental axon pruning corrects any alterations in topographic axon order in the CC, these deficits persist in PND25 AnkB440 KO mice, and together with axon defasciculation (see below) likely contribute to the volumetric tract changes we observe.

Could CC thickness increase be due to axon-axon defasciculation? L1-CAM- Nrp1 labeling could be achieved to look at how fibers are organized in the CC tract.

We thank the reviewer for raising this question. Our new data provides evidence of axon defasciculation.

PND25 brains stained with L1CAM or sparsely labeled axon tracts using the Thy1-GFP-M reporter showed changes in the size and organization of multiple axon tracts in AnkB440 KO brains and the presence of both defasciculated and mistargeted axons (Figure 2).

Is it feasible to distinguish the response of the growth cone of the main axon from that of the growth cones from the collaterals? Do the authors think that the lack of response to Sema3A of ANKB440 cortical neurons will impact in vivo on the pruning or rather on the initial number of formed collaterals?

As we show in Figure 6, the response to Sema 3A is affected in growth cones of both the primary axon and collateral branches in AnkB440 KO neurons, which in turn likely affects collateral branch pruning in vivo. As we now show, PND25 AnkB440 KO brains also have deficits in the topographic ordering of callosal axons originating from S1 in the CC, and in the targeting and developmental refinement of their contralateral projections (Figure 4), defects also observed in mice with impaired Sema 3A signaling (Zhou et al. 2013; Martín-Fernández et al., 2021). These findings support an involvement of AnkB440 in Sema 3A-mediated callosal connectivity and that deficits in developmental axon pruning contribute to the altered tract development in AnkB440 KO brains.

Our in vitro data also offers a rationale for a higher number of initial collaterals based on the higher number of actin patches that form in AnkB440 KO neurons, which correlates with more branch initiation points, which together with the previously reported mechanism of higher propensity of microtubule invasion of nascent branches and subsequent branch stabilization, are likely to also promote axon hyperbranching. Our life imaging data also suggests that loss of AnkB440 diminishes the inhibitory effects of Sema 3A in collateral branch formation (Dent et al., 2004).

-Given the defective response to a guidance cue, cortical axons could select wrong trajectories. Do the authors think this is the case? This could be examined at developmental stages with DiI tracing.

Our Dil tracing attempts yielded inconsistent labeling across specimens. On the other hand, GFP labeling of axon tracts via IUE or the Thy1-GFP-M reporter revealed aberrant innervation of axons in regions of the septofimbrial nucleus and the fornix (Figure 2C) as well as increases in contralateral subcerebral projections (Figure 4B) in PND25 AnkB440 KO brains, which are indicative of axon misrouting.

– Is there a reason thinking that defects in the mutant are restricted to the corpus callosum?

We thank the reviewer for raising this important question. We have included new data showing that changes in axon tract morphology is not restricted to the CC. As mentioned above, we sparsely labeled a subset of axon tracts using the Th1-GFP-M reporter and detected volumetric deficits in the anterior commissure, the ventral hippocampal commissure, and the septofimbrial nucleus (Figure 2C-E).

– Is ANKB440 deficit impacts on the early polarity of cortical neurons. In culture, do the neurons extend similar number of processes before axon specification? Is the observed difference of F-actin patches in context of ANKB440 deletion expected to impact on process initiation?

We have previously shown that combined loss of AnkB isoforms does not impact axon specification or axon polarity (Lorenzo et al., 2014). We have now added new data (Figure 1—figure supplement 2B,C) that show that loss of AnkB440 does not change the number of neurites prior to axonal specification. On the other hand, as we discussed in the original submission, loss of AnkB440 results in higher number of actin patches, which correlate with a larger number of branch initiation points, nascent filopodia, and collateral branches. As we discuss in the manuscript, we do not understand how AnkB440 regulates the formation of actin patches and its other effects on actin dynamics, a question we plan to investigate in future studies.

– How could they explain the maintained sensitivity to EphrinA5, since it was reported that L1-CAM is required for the response to EphrinA5 (Demyanenko et al., 2011)?

Demyanenko et al., 2011 observed a modest (18%) reduction in the growth cone collapse in L1CAM null cultured neurons in response to Ephrin A5 relative to control neurons. We have added new data (Figure 5— figure supplement 2C,D) showing that, like the AnkB440 KO neurons, mutant Y1229H L1CAM cortical neurons, in which the AnkB440-L1CAM association is disrupted, respond normally to Ephrin A5. It is possible that to elicit a noticeable difference in L1CAM-mediated GC collapse in response to Ephrin A5, a total loss of L1CAM is required, and this mechanism does not seem to be transduced through AnkB440.

– Do the authors think that alteration of active/inactive cofilin ratio could have additional outcome in the mutant?

Given the wide range of neuronal process involving actin dynamics, this is certainly an intriguing question worth examining in future studies. As we speculate in the manuscript, the increases in active cofilin in AnkB440 KO mice during early brain development might provide a more dynamic actin pool and could underlie the surges in axonal actin patches and emerging filopodia observed in AnkB440 KO cortical neurons.

– The authors should indicate more clearly in the manuscript, or figures the number of replicates, the number of analyzed cases for each experiment. They should also indicate in the figure panels the developmental stage at which the observations were made.

Thank you for raising this point. The revised manuscript includes the previously missing details about the experimental conditions pointed out by the reviewer.

Reviewer #3 (Recommendations for the authors):The interpretation of the results in Figure 2A/B are a bit confusing. Do the authors suggest that the corpus callosum is thicker due to branching before the corpus callosum? In Figure 2E/F they show increased branching near the cell bodies on the ipsilateral side of the cortex, but how would this relate to the callosal defect? Since the difference in callosal thickness is only close to the midline, wouldn't this suggest some defect is specifically occurring there?

We ruled out that morphometric changes in the CC in AnkB440 KO brains are due to non-cell-autonomous effects such as increases in the proliferation and maturation of oligodendrocytes, or in myelination (levels and wrapping) of CC axons. On the other hand, we have detected changes in axon fasciculation, which may be a contributing factor to the increase in CC thickness. As discussed above (see first response to reviewer 2), we also observe an increase in the number of callosal processes at the midline of PND25 AnkB440 KO brains (Figure 4). This increase is accompanied by a more widespread dorsoventral distribution of S1 callosal axons, with a significant number of axons navigating through more dorsal CC paths. Thus, CC hyperplasia in PND25 AnkB440 KO brains could also result from a higher density of axons entering and traveling through the CC and/or from deficits in the ordering and developmental pruning of CC axons. Our results indicate deficits in the removal of the vast majority of S1 callosal axons that travel throughout the dorsal areas of the CC at the brain midline, which are also observed in mice deficient in Sema 3A signaling (Zhou et al. 2013; Martín-Fernández et al., 2021). Increases in the number of CC axons in AnkB440 brains could reflect misrouted axons or deficient postnatal elimination of transient callosal axons derived from S1 layer IV neurons (De León Reyes et al., 2019, https://doi.org/10.1038/s41467-019-12495-w). Lastly, it is also possible that AnkB440 KO axons respond abnormally to other midline cues.

Figure 3C – The middle two images (and labels) in the top row of images should be switched to be consistent with the bottom two rows.

Thank you for catching this error. This is now fixed.

The title of Figure 3 is that AnkB440 is enriched at axonal growth cones. This is an overinterpretation. The figure shows that AnkB440 is present in growth cones but it does not appear to be enriched there. To say it is enriched there they would have to show more convincing images and quantify the levels in the growth cone and axon shaft (normalized to cytoplasmic protein) and show enrichment.

We have replaced “enriched” with “abundantly expressed” or similar phrasing, which more clearly captures our point about the expression of AnkB440 in the growth cone.

Rescue of the AnkB440 KO phenotype is shown in Figure 4E-G with AnkB440-Halo but not AnkB220-GFP, which is a nice experiment. It shows AnkB440 is required for the Sem3A phenotype and it is specific to the neural-specific isoform. Why weren't these experiments carried out for the filopodia and branching experiments in Figure 1D-H and Figure 1 – figure supp. 2?

The revised manuscript now includes rescue experiments with AnkB220 and AnkB440 cDNAs for actin patches and filopodia (Figure 1 D-H and Figure 1—figure supplement 2A) and collateral branches (Figure 1—figure supplement 3A,B).

Figure 4 B,C and D should not be presented as line graphs, as they were not repeated measures but separate experiments for each treatment time.

Thank you for suggestion this correction. These data are now presented as graph bars for the individual time points.

The experiment in Figure 4 – Figure Supp. 4 with Ephrin-A5 still collapsing AnkB440 KO growth cones shows nice specificity to the lack of response to Sema3A and convincingly indicates the growth cones can collapse, indicating a specific, rather than general defect in growth cone response.

We agree. Please see response to comment #6 from Reviewer 2 for additional commentary about this result.

Figure 5B – The horizontal and vertical labels appear to be reversed based on the images of the growth cones.

Thank you for catching this error. This is now fixed in Figure 6B.

It is unclear why an unpaired t-test was used in 4G but ANOVA was used in Figure 5C. ANOVA is appropriate for multiple conditions, t-tests are not. Unpaired t-test shouldn't be used in Figure 6B, D, Figure 6-Figure supp. 6, Figure 8F – for multiple conditions.

Thank you for catching this error. This is now fixed in Figure 5G, Figure 5—figure supplement 2B and Figure 9D,F. Please note that we used t-test for the statistical comparisons in now Figure 7B,D because those graphs represent the percent expression of each individual protein relative to their levels in the control group. We are independently comparing levels of each protein (each bar in the graphs) between two groups, but not changes among different proteins (bars) or how they relate to each other.

The keys are confusing in several of the figures. In Figure 2D the dots for the AnkB440 KO are one color in the key but different in the graph. In Figure 4G the bars are in the key, not the dots. In Figures 5C/E, 6H and 7E the key is again the color of the dots, not the bars. Also, the dots for AnkB440 KO and L1CAM-Y1229H are very similar color, making it confusing. To make it easier to interpret, the bar color, not the dot color, should be used in the keys.

We have adopted the suggestions of the reviewer and made the corresponding changes in all figures of the revised manuscript.

Could the authors include a working model at the end? How is AnkB440 functioning with L1CAM, Nrp1 and cofilin to influence the actin cytoskeleton in response to Sema3A? The authors have put together many very nice experiments but in the discussion it was hard to put together a mechanistic model of how AnkB440 is functioning in growth cone collapse/branch inhibition.

We have included a model depicting the AnkB/L1CAM/Nrp1 complex as receptor and transducer of Sema 3A guidance cues at the surface of GCs to facilitate the transduction of these signals to the actin cytoskeleton. Local changes in cofilin phosphorylation downstream of this Sema 3A pathway induce F-actin disassembly and GC collapse (Figure 10).